# Maternal anxiety affects embryo implantation via impairing adrenergic receptor signaling in decidual cells

Jinxiang Wu[1✉], Shu Lin[2,3], Pinxiu Huang[4], Lingling Qiu[1], Yufei Jiang[4], Ying Zhang[4], Nan Meng[4], Meiqing Meng[4], Lemeng Wang[4], Wenbo Deng[4], Zhao Liu[4], Chuanhui Guo[4], Jinhua Lu[4], Haibin Wang[4✉] & Shuangbo Kong[iD][4✉]

Recurrent implantation failure (RIF) is defined as the failed pregnancy after good embryo transfer over 3 cycles during in vitro fertilization (IVF).The human endometrium plays a vital role in providing the site for embryo implantation, with several factors implicated in unsatisfactory endometrial receptivity in RIF. Our present results revealed that women with pregnancy loss or infertility have a higher serum epinephrine level, indicating a potential correlation between psychological stress and pregnancy failure. RNA-sequencing of the tissues collected at the endometrial receptive phase in normal and RIF women showed that stress hormones could affect the functional status of endometrial receptivity. Subsequent analysis revealed that the epinephrine signaling acts as an important regulator of endometrial receptivity through the PI3K-AKT and FOXO1 signaling pathways. We also found that patients with RIF show attenuated expression of the alpha-2C-adrenergic receptor (ADRA2C) and that its down regulation induced by high level epinephrine could inhibit the decidualization. Early pregnant mice treated with stress showed high serum epinephrine levels, defective uterine adrenergic receptor expression, and low pregnancy rates. Altogether, our findings indicate that mental stress during early pregnancy can alter the functional status of endometrial receptivity.

[1] Department of Reproductive Medicine, the Second Affiliated Hospital of Fujian Medical University, Quanzhou, Fujian Province, China. [2] Centre of Neurological and Metabolic Research, the Second Affiliated Hospital of Fujian Medical University, Quanzhou, Fujian Province, China. [3] Diabetes and Metabolism Division, Garvan Institute of Medical Research, 384 Victoria Street, Darlinghurst, Sydney, NSW 2010, Australia. [4] Fujian Provincial Key Laboratory of Reproductive Health Research, Department of Obstetrics and Gynecology, The First Affiliated Hospital of Xiamen University, School of Medicine, Xiamen University, Xiamen, Fujian, China. ✉email: pursuer@163.com; haibin.wang@vip.163.com; shuangbo_kong@163.com

Psychological anxiety in women of childbearing age is one of the most common reasons for infertility. Assisted reproductive technology has made some progress in the treatment of infertility. However, approximately 50% of the pregnancies continue to fail clinically despite using a good quality of embryo for transfer[1,2]. The human endometrium plays a vital role in providing the position for embryo implantation. Hence, unreceptive endometrium and abnormal early maternal–fetal interaction may lead to infertility, including repeated failure of embryo implantation. These adverse pregnancy conditions can affect the quality of life of these couples as well as their psychological state.

The human endometrium is a steroid-responsive tissue that undergoes a series of regeneration cycles, including sequential proliferation, differentiation, destruction, and repair. These changes constitute together the menstrual cycle[3]. This dynamic tissue remodeling system ensures that the endometrium can be successfully implanted in a short time—the so-called "implantation window". During the blastocyst implantation step, the endometrial stroma embeds the invasive embryos. During this time, the endometrial stromal cells undergo extensive differentiation and revascularization and recruit immune cells. This process is called "decidualization", and it is pertinent for inducing a successful pregnancy[4]. Decidualization of HESCs requires eliciting the correct response to changes in the peripheral hormones. The degree of decidual reaction in different species is related to the depth of the placenta, suggesting that the decidual process dominates the invasion of trophoblast cells[5,6]. In humans, decidualization is initiated regardless of pregnancy status, and it is mainly a response to ovarian steroids, such as progesterone, in the middle of each luteal phase[7,8].

Recurrent implantation failure (RIF) and recurrent pregnancy loss[9] are common adverse reproductive events occurring during assisted reproduction. Epidemiological data have shown that patients with early adverse pregnancy outcomes suffer from mental tension and anxiety[10,11]. Maternal stress during early pregnancy is strongly associated with various complications occurring during an ongoing gestation period[12]. Several non-randomized control trials have reported that psychological support in early pregnancy reduces the rate of miscarriage in women with unexplained miscarriage(s)[13,14]. However, only a few studies have focused on the underlying mechanisms of psychological stress associated with reproductive health, with only a handful of investigative studies conducted so far. In a study, the plasma epinephrine and norepinephrine levels of women with RIF were observed to be significantly higher than those of the women in the control group[15]. Our previous study showed that maternal epinephrine exposure during early pregnancy impairs the process of uterine decidualization and embryo development in mice[16]. In the present study, we conducted further research and have found that serum epinephrine levels are closely related to infertility and embryo implantation failure. In addition, we confirmed that the ADRA2C was highly expressed in the decidual cells during the "implantation window", which plays vital physiological functions. Herein, we revealed the mechanism of psychological stress undermining the endometrial functions and embryo implantation.

## Results

**Serum epinephrine and norepinephrine were significantly elevated in women with infertility and RIF.** Considering the strong association between mental stress and infertility, we attempted to test whether the serum levels of stress-related hormones differed in infertile women. Since epinephrine and norepinephrine were crucial stress hormones[17,18], we tested the levels of these hormones in patients with and without infertility. The levels were found to be significantly higher in women with infertility than in the corresponding controls (Fig. 1a, b). However, in patients with infertility who underwent embryo transfer, only the serum epinephrine levels of those with failed embryo implantation were significantly higher than those with pregnancy (Fig. 1c, d). Since all patients were transferred with good-quality embryos, human endometrial receptivity dysfunction could have been the main cause of pregnancy failure.

**Aberrant endometrial gene expression in RIF women during the receptive phase.** The analysis of the transcriptome data from seven endometrial samples with RIF and six control samples revealed that 2384 mRNAs were differentially expressed (1115 downregulated and 1269 upregulated, log2-fold change ≥ 0.0 and P value ≤ 0.05 as the cutoff criteria) (Fig. 1e). The PCA plot of the samples is presented in Supplementary Fig. S1.These results demonstrated that the gene profiles of the endometrium with RIF were significantly different from those of the corresponding control. Next, we applied the Gene Ontology (GO) analysis to predict the functions of aberrantly expressed genes. Of these DEGs, 378 mRNAs were downregulated, while 240 mRNAs were upregulated, with |log2-fold change| ≥ 1.0 and P value ≤ 0.05 serving as the cutoff criteria. The biological processes of GO is depicted in Fig. 1f. The upregulated genes were mainly involved in the translation and cell proliferation, while the downregulated genes were associated with immune regulation and inflammatory response. Immune and inflammatory signals were confirmed to be important for embryo implantation and early pregnancy[19,20] (Fig. 1f).

**The adrenergic receptor ADRA2C was highly expressed in decidualized HESCs.** We then assessed whether adrenergic receptor signaling was involved in human endometrial receptivity. We accordingly generated a scatter plot to reveal the transcriptomic patterns of adrenergic receptors (Fig. 2a) and found that only *ADRA2C* was highly expressed in the mid-secretory phase, albeit it decreased in RIF (Fig. 2a). We further checked the expression of ADRA2C in the human endometrium and found that it was significantly enhanced during the endometrial secretory phase (Fig. 2c, d). While investigating the cellular localization of ADRA2C in human endometrium, as shown in Fig. 2b, staining for ADRA2C was low or undetectable in the proliferative specimens, but positive in the secretory stage. We also found that *ADRA2C* was highly expressed in the glandular epithelial cells, luminal epithelial cells, and stromal cells in the receptive endometrium.

Since the ADRA2C receptor was highly expressed in stroma cells and decidualization of HESCs was deemed crucial for the establishment of endometrial receptivity, we utilized an immortalized human endometrial stromal cell line, which could be induced to decidualization, so as to employ the function of adrenergic signal for the endometrium[16]. We screened the expression profiles of different adrenergic receptors by RT-PCR during decidualization, and only *ADRA2C* exhibited an enhanced expression pattern during the in vitro decidualization (Fig. 2e, f). The expression of ADRA2C protein was enhanced with the differentiation of HESCs (Fig. 2g, h). Moreover, we detected *phenylethanolamine N-methyltransferase* (*PNMT*) and *tyrosine hydroxylase* (*TH*), which were the key rate-limiting enzymes for the synthesis of adrenergic ligands[21,22]. These results signified that the adrenergic receptors were present in the human endometrium; however, HESCs could not synthesize epinephrine ligand (Supplementary Fig. S2a, b).

**Adrenergic signaling was involved in the decidualization of HESCs.** To elucidate the role of ADRA2C in the decidualization of HESCs, we examined the expression level of classical decidualization markers, including *prolactin* (*PRL*) and *insulin-like*

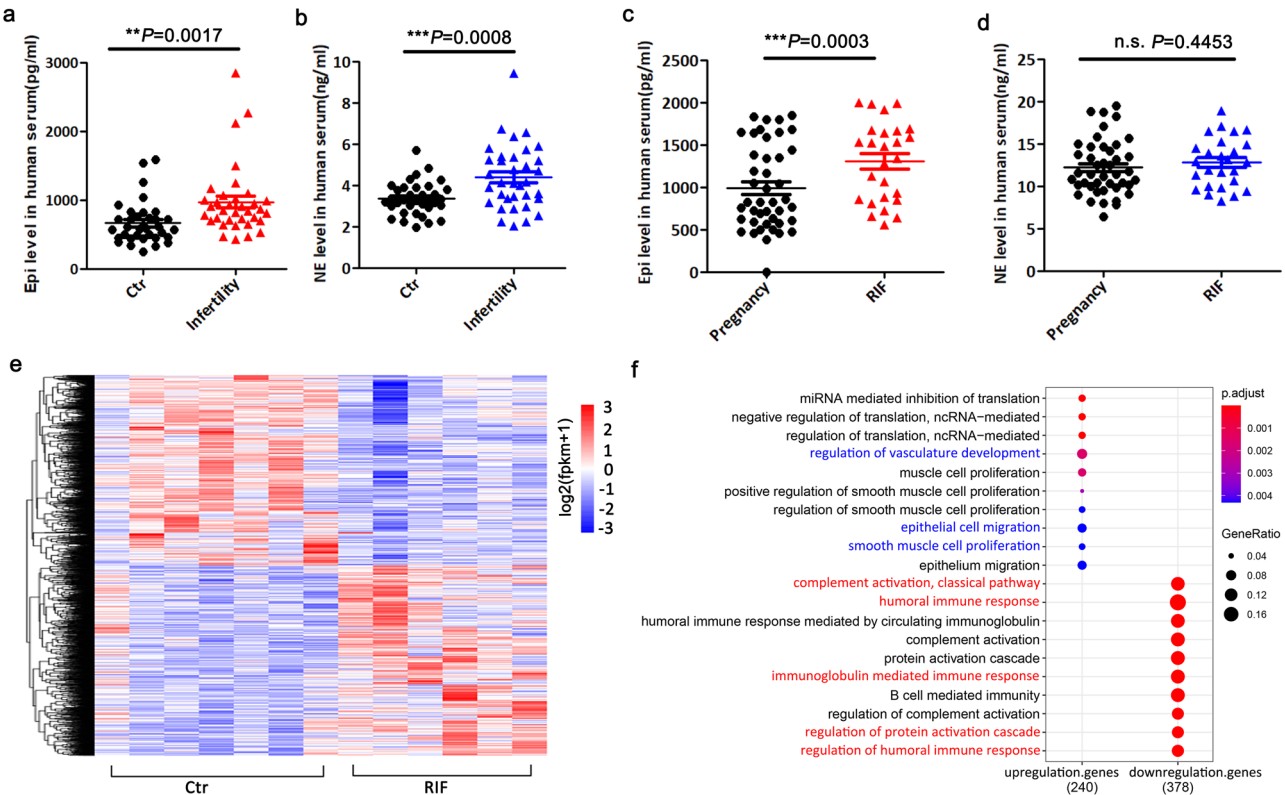

**Fig. 1 The changes in serum epinephrine and norepinephrine levels and transcriptomic-wide differences for endometrium in RIF patients through RNA-seq analysis. a, b** Serum epinephrine and norepinephrine levels in control patients ($n = 35$) and infertility patients ($n = 35$); Ctr control, Epi epinephrine, NE norepinephrine. **c, d** Serum epinephrine and norepinephrine levels in pregnancy patients ($n = 43$) and RIF patients ($n = 26$); RIF recurrent implantation failure. **e** Heatmap of upregulated and downregulated endometrial genes in the putitive recetive windows in RIF compared with control (upregulated genes in red and downregulated genes in blue); 1269 genes were upregulated, 1115 genes were downregulated. Significance was indicated as -log10 $p$ value, $p$ value $\leq 0.05$ and |log2FoldChange| $\geq 0.0$. **f** Significantly enriched GO terms for the DEGs in receptivity endometrium with RIF (upregulated genes in red and downregulated genes in blue). Values are represented as the mean ± SEM. For statistical analysis, $p$ values were calculated by one-way ANOVA followed by Tukey's multiple comparison post hoc test. Statistical significance for all analyses were *$p < 0.05$; **$p < 0.01$; ***$p < 0.001$.

growth factor-binding protein 1 (*IGFBP-1*), in different concentrations of epinephrine-treated HESCs. We found that only 5.0 μM epinephrine could effectively promote decidualization (Fig. 3a, b) and that this promotion was time-dependent (Fig. 3c, d). In addition, adrenaline loss of 10.0 μM had promoting effects relative to that of 5.0 μM (Fig. 3a, b, e, f). Since *ADRA2C* was weakly expressed in undifferentiated HESCs, we used 0–10.0 μM epinephrine for cell growth analysis in order to evaluate its effect on HESCs culture in vitro. As shown in Supplementary Fig. S3, different concentrations of epinephrine displayed little effect on the proliferation activity of HESCs. Therefore, we used the exposure dose of 5.0 μM epinephrine for further studies.

**Adrenergic signaling inhibited AKT activity to increase the Forkhead box O1 (FOXO1) expression and nuclear localization.** The observation that an appropriate level of adrenergic signal promoted the HESCs decidualization motivated us to explore the underlying mechanism for ADRA2C regulating the normal process of human endometrial stromal differentiation. ADRA2C belongs to G-protein-coupled receptors, and its common downstream activation pathways include PKA, ERK, and AKT[23]. Western blotting was conducted to detect the expression level of these signaling molecules. Only the AKT signaling pathway was affected by epinephrine stimulation, and the phosphorylation level of p-AKT (Ser473) was reduced by 5.0 μM epinephrine stimulation, but not by 10.0 μM epinephrine stimulation (Fig. 4a, b). Moreover, we also noticed that the ADRA2C receptor was downregulated in the high

(10.0 μM) epinephrine-treated group (Fig. 4a, b). A wealth of data indicated that AKT pathway activity was involved in decreased FOXO1 protein and that it blunted the decidual response[24,25]. Furthermore, as expected, the level of FOXO1 protein increased with a decrease in the AKT phosphorylation level (Fig. 4a, c). We also assayed other transcription factors such as HOXA10 and STAT3, which were the key mediators in decidualization[26–28]. Interestingly, the protein level of factor HOXA10 was increased by epinephrine stimulation, but not by epinephrine concentration and the protein level of ADRA2C (Fig. 4c, d). We found that the change of FOXO1 was affected by epinephrine concentration; moreover, these changes were affected by AKT phosphorylation signal regulation (Fig. 4a, c). In addition, 10.0 μM epinephrine did not promote the decidualization as much as that by 5.0 μM epinephrine, which was consistent with the report of its inability to increase the FOXO1 level. Based on these results, we suggested that the physiological level of epinephrine was beneficial for decidualization through its receptor ADRA2C, and aberrantly elevated epinephrine loss of its promoting effects for decidualization. During decidualization, FOXO1 was responsible for transcribing the *IGFBP-1* and *PRL*[29,30]. FOXO1 might be compartmentalized in both the cytoplasm and nucleus. Nonetheless, it must be located in the nucleus to perform the transcriptional activity. Hence, we next performed Western blotting and immunofluorescence experiments to detect the distribution of FOXO1 in the cytoplasm and nucleus of HESCs. Indeed, the nuclear localization of FOXO1 was increased during HESC differentiation under the induction of 5.0 μM

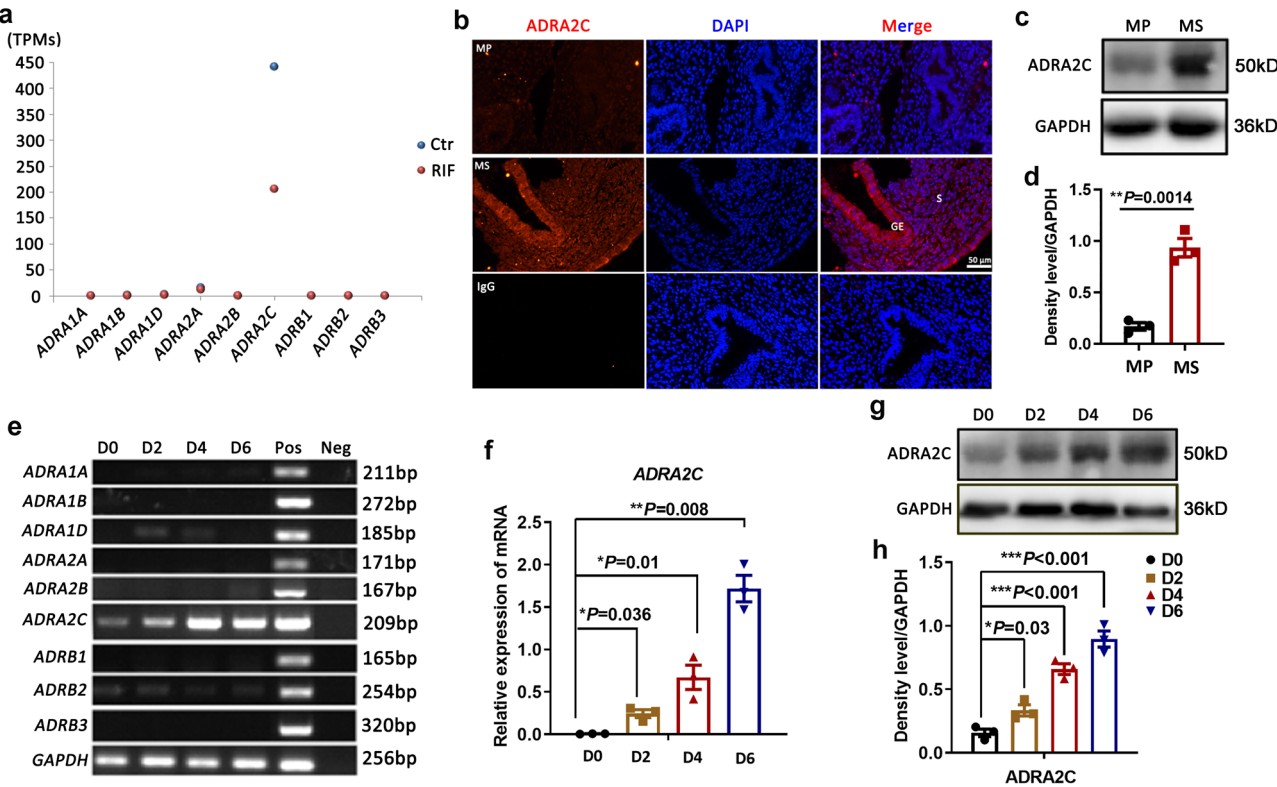

**Fig. 2 ADRA2C was robustly expressed in HESCs during decidualization. a** Analysis of the alterations transcriptomic patterns of adrenergic receptors in RIF versus control women. TPM transcripts per million. **b** Tissue immunofluorescence for ADRA2C protein location in the Mid-proliferative stage and Mid-secretory phase of human endometrium. Scale bars, 50 μm. MP mid-proliferative phase (day 8 of the menstrual cycle), MS mid-secretory phase (day LH +7), S stroma, GE glandular epithelium, IgG for control. **c** Western blotting for the ADRA2C protein level between mid-proliferative and mid-secretory phase of endometrial tissues. **d** Relative fold change in density in **c**. **e** RT-PCR screen for the expression of different adrenergic receptors during decidualization of HESCs. Pos positive control, Neg negative control. **f** Quantitative RT-PCR analysis of *ADRA2C* mRNA level in HESCs during decidualization. **g** Western blot for ADRA2C protein expression before and after HESCs differentiation. D0, HESCs cultured in vitro before differentiation; D4, HESCs induced differentiation on day 4 in vitro. **h** Relative fold change of density levels for **g**.

epinephrine, but not by 10.0 μM epinephrine (Fig. 4e, g). Collectively, these results suggested that ADRA2C activation was an important process in assisting the decidualization of HESCs and that high concentrations of epinephrine restrain the process of HESCs decidualization.

**Aberrant ADRA2C signaling hampered HESCs decidualization.** To verify whether the clearance of the accumulated ADRA2C reversed the effects of assisting HESCs decidualization, we employed a small interference RNA (siRNA) technique to knock down the expression of *ADRA2C* in HESCs. As shown in Fig. 5a–c and Supplementary Fig. S4a, during the decidualization process, ADRA2C-siRNA largely reduced the *ADRA2C* expression at both the mRNA and protein levels in the HESCs. This decreased expression could partially eliminate the synergistic effect of adrenergic signals on the decidualization of HESCs. Real-time quantitative PCR was applied to evaluate the decidualization markers *IGFBP-1* and *PRL* (Fig. 5d, e). Consistent with this observation, the knockdown of *ADRA2C* expression largely enhanced the activity of AKT signaling relative to that in the control group (Fig. 5f, g). This increase significantly lowered the level of FOXO1 protein, which attenuated the effects of adrenergic signaling to promote the decidualization of HESCs (Fig. 5f, g). However, the AKT activity and FOXO1 level did not change in the absence of epinephrine stimulation after the knockdown of ADRA2C (Fig. 5h, i). These results further demonstrate that epinephrine could inhibit the AKT activity and enhance the FOXO1 level by stimulating ADRA2C in HESCs decidualization.

Then, we tested whether *ADRA2C* knockdown would reduce the nuclear localization of FOXO1. As shown in Fig.5j and Supplementary Fig. S4b, when compared with the control group, the level of FOXO1 protein and the localization of FOXO1 in the nucleus of HESCs were both reduced after the *ADRA2C* knockdown. These findings further strengthen the notion that ADRA2C signaling and HESCs decidualization are functionally associated. Overall, the results of these experiments suggest that epinephrine could promote HESC decidualization by inhibiting the AKT activity to increase the FOXO1 expression and the nuclear localization via ADRA2C (Fig. 5i).

**High epinephrine compromised the function of decidual cells and embryo development in mice.** To further verify the detrimental effects of a high level of adrenergic ligand on decidualization and early pregnancy in vivo, a mouse stressed model and exogenous epinephrine exposure mouse model were used in this study (Fig. 6a, e). Stress-induced mice showed increased serum epinephrine levels (Fig. 6b, c) but significantly decreased mating rate (Fig. 6d). Exposure to both endogenous and exogenous epinephrine in mice resulted in poor embryonic development (Fig. 6I, j). As shown in Fig. 6i, the arrow indicates the growth-restricted implantation sites on D6 and D8 of the stressed mouse models. We used an artificially induced decidualization mouse model (Fig. 6f) and found that exogenous epinephrine exposure could impair the decidual differentiation of endometrial stromal cells (Fig. 6g, h). This result indicated that the significantly decreased expression of these receptors was not caused

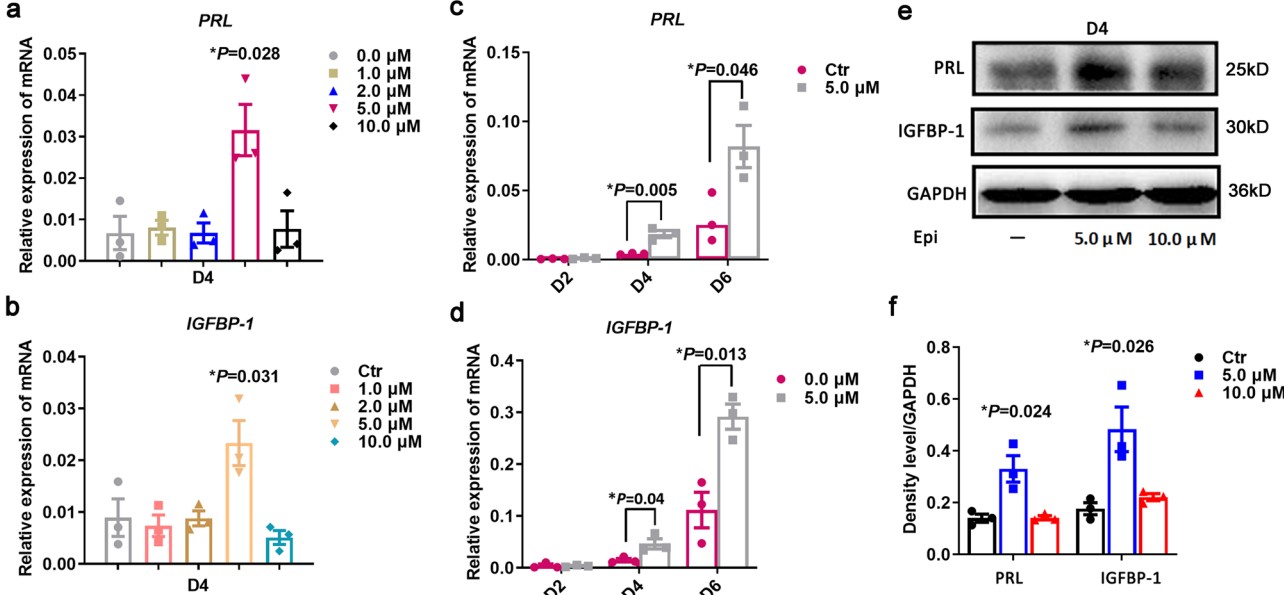

**Fig. 3 Dose-dependent role of adrenergic signaling in HESCs during decidualization. a, b** The mRNA levels of decidualization markers *prolactin* (*PRL*) and *insulin-like growth factor-binding protein 1* (*IGFBP-1*) on the 4th day of decidualization in the control and epinephrine-treated group. The values were normalized to the *GAPDH* expression level and indicated as the mean ± SEM, $n = 3$. **c, d** The mRNA levels of decidualization markers *PRL* and *IGFBP-1* in control and 5.0 μM epinephrine treated HESCs during induced decidualization at different time. The values were normalized to the *GAPDH* expression level and indicated as the mean ± SEM, $n = 3$. **e** Western blot for PRL and IGFBP-1 protein expression level, with GAPDH as a loading control. **f** Relative fold change in density levels for **e**.

by abnormal embryonic development. These interesting findings suggest that elevated epinephrine levels in vivo disrupt the decidualization of uterine stromal cells, leading to early embryo development.

**High levels of epinephrine in vivo may affect the endometrial implantation window by downregulating the adrenoreceptor signaling.** Our past study found that, when compared with the endometrial proliferation phase, the level of ADRA2C protein in the endometrial secretory phase was increased. Moreover, we found that an appropriate dose of epinephrine could promote the differentiation of HESCs (Fig. 3). Interestingly, we discovered that high concentrations of epinephrine did not promote the differentiation of HESCs in vitro, but that it downregulated the expression of its receptor ADRA2C (Fig. 4c). Since the serum epinephrine level of the RIF group was higher than that of the control group (Fig. 1c), we examined whether endometrium from the mid-luteal phase of the menstrual cycle displayed a similar alternation as that observed in vitro. Primarily, we found that mice with high serum epinephrine levels corresponded to higher epinephrine in the uterine tissues when compared to the controls (Fig. 7a). This means that the level of circulating epinephrine fluctuates in a similar way to the level of epinephrine in the uterine tissue. We speculated that these impairments of decidualization may be related to abnormal adrenergic receptor signaling. Our previous studies confirmed that mouse decidual cells in early pregnancy expressed *Adra1b*, *Adra2b*, and *Adrb2* receptors[16]. We then validated the expression of these receptors in a mouse model and found that both endogenous and exogenous epinephrine reduced the expression of adrenergic receptors in the uterine decidual cells during early pregnancy (Fig. 7b–d). Moreover, through in situ detection, we found that the expressions of these adrenergic receptors in the decidual cells of the stressed mouse and exogenous epinephrine exposure mouse were significantly reduced (Fig. 7e, f). We also identified that *ADRA2C* was involved in significantly differentially

downregulated genes, as displayed in the volcano map (Fig. 7g). WB data showed that the protein level of ADRA2C in the mid-luteal phase of the menstrual cycle from RIF patients was significantly lower than that of the controls (Fig. 7h, i). In addition, we performed pathway analyses to identify the significant pathways associated with the DEGs according to the Kyoto Encyclopedia of Genes and Genomes (KEGG) (Fig. 7j). The KEGG pathway analysis revealed that Akt signaling was involved in the affected pathway among the upregulated genes (Fig. 7j). Therefore, these findings were consistent with the in vitro cellular results. We hypothesized that the serum epinephrine levels possibly alter the endometrial receptivity by affecting the expression of adrenal receptors; thus, we believed that the downstream signal cascade involved AKT-FOXO1 in the endometrium.

## Discussion

The main study findings are as follows: increased serum epinephrine levels are closely related to infertility and failed/defective embryo implantation; during the "implantation window," epinephrine-receptor ADRA2C is highly expressed in the stromal or decidual cells; appropriate activation of ADRA2C signal can reduce the phosphorylation level of AKT and upregulate the level of FOXO1 protein and its nuclear localization, thereby promoting the decidualization of endometrial stromal cells; the serum epinephrine level was higher and the pregnancy rate was lower in stressed mice, which displayed a defective expression of uterine adrenergic receptor and abnormal decidua. In short, by multiple approaches, the molecular mechanism was uncovered to show that psychological stress directly disrupted endometrial functioning and early embryo implantation.

A recent study reported that exogenous epinephrine and norepinephrine could affect the development of mouse uterine decidua and early embryonic development[16,31]. These studies have confirmed that excessive exogenous epinephrine can cause signal abnormalities during decidual development, including *Ptgs2* (encoding COX2), *Wnt4*, and *Bmp2* signaling. In addition,

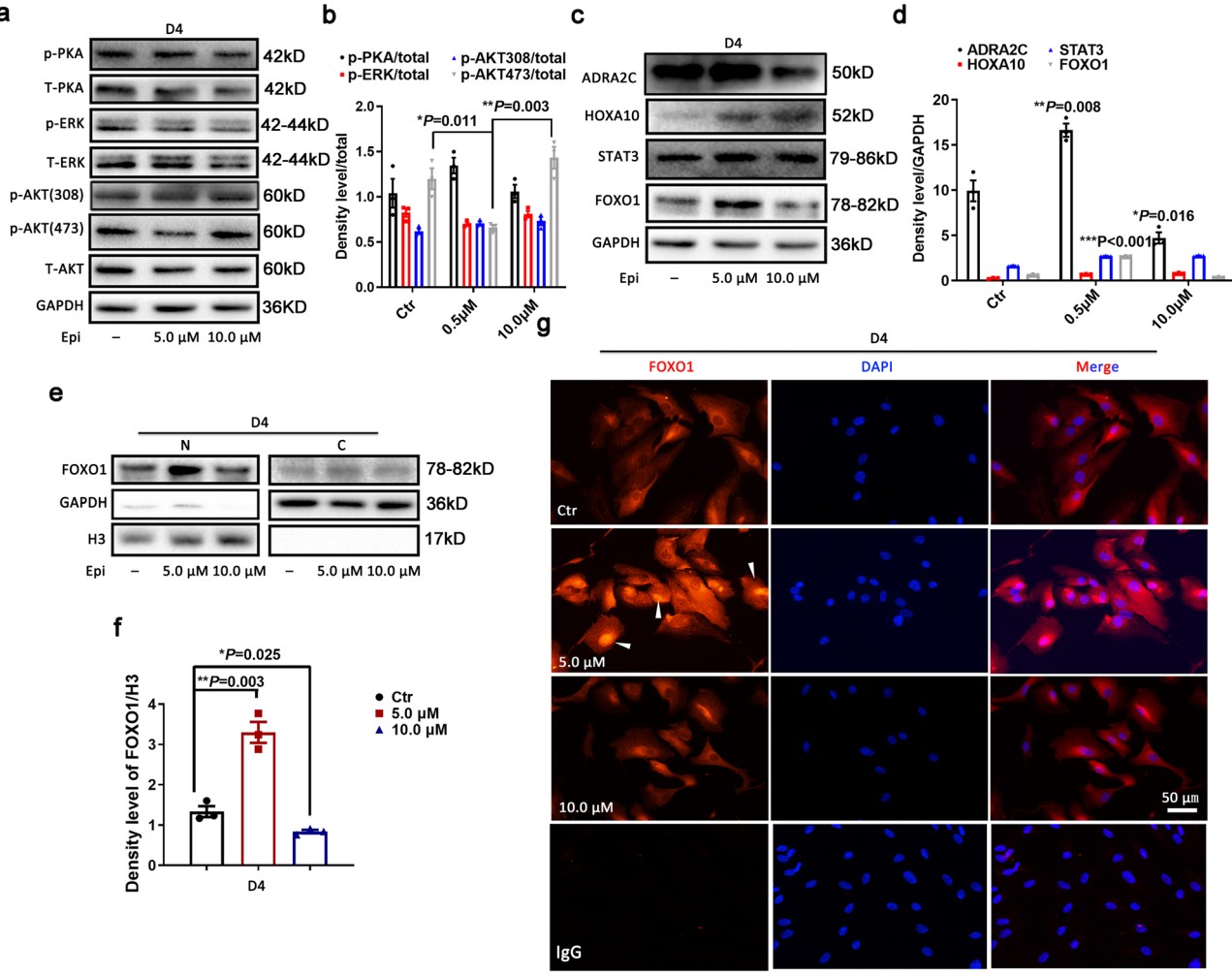

**Fig. 4 The adrenergic signaling pathway downregulates AKT activity, associated with increased FOXO1 expression and nuclear localization. a** Western blot analysis of total protein and active forms of PKA, ERK, and AKT signaling pathways in D4 decidualized cell with different treatments. **b** Relative fold change in density levels for **a**. **c** Western blot analysis of ADRA2C and decidual development-related transcription factors HOXA1O, STAT3, and FOXO1. **d** Relative fold change in density levels for **c**. **e** Western blot analysis of FOXO1 in the nucleus and cytoplasm fraction. N nucleus, C cytoplasm. **f** Relative fold change in density levels for **e**. **g** Expression and localization of FOXO1 by cellular immunofluorescence assay. The arrow points out FOXO1 in the nucleus. Scale bars, 50 μM. IgG for control.

several critical transcriptional factors during decidualization, including FOXO1, Meis Homeobox 1 (MEIS1), and CCAAT Enhancer Binding Protein Beta (CEBPB) were decreased (Fig. 8). Exogenous epinephrine exposure during the early pregnancy resulted in impaired decidualization and early pregnancy in mice[16]. RNA-Seq data showed that ADRA2C was highly expressed in the human endometrium, signifying the underlying physiological function of adrenergic receptor signaling in the human endometrium. Furthermore, we observed that ADRA2C signaling was highly present in both the endometrial epithelia and stroma, and that the expression was lost in both the compartments of RIF patients. This observation suggested that adrenergic receptor signaling might exhibit biological functions in both epithelial and stromal differentiation. Decidualization refers to the transformation of endometrial stromal fibroblasts into characteristic secretory decidual cells that provide nourishment and an immune-privileged matrix for embryo implantation and placental development[32]. Despite the clinical importance of decidualization, the underlying mechanisms remain to be elucidated in vivo owing to the related complexity. Only a few studies have so far investigated whether neuropsychiatric factors are involved in the establishment of a functional endometrium, especially

during endometrial stromal cell decidual differentiation. In the present work, we confirmed the presence of adrenergic neurotransmitters and their receptors in the endometrium and demonstrated their physiological regulatory significance in establishing endometrial receptivity.

From a mechanistic perspective, we detected the decidualization of endometrial stromal cells and the gradual increase in the ADRA2C expression, which resulted in a decrease in the level of phosphorylated AKT and thus the FOXO1 nuclear translocation. The more nuclear-localized FOXO1 leads to an increase in the *IGFBP-1* and *PRL* mRNA levels under epinephrine stimulation (Fig. 4). As mentioned earlier, FOXO1 is a transcription factor that plays an important role in the process of decidualization, which is necessary for endometrial receptivity and for regulating the expression of genes encoding *IGFPB-1* and *PRL*[30,32]. In mammalian cells, AKT phosphorylates Ser256 to inhibit FOXO1 activity, leading to its nuclear export and transcriptional inactivation[33]. Moreover, it has been demonstrated that FOXO1 is dephosphorylated during phosphatidic acid-induced decidualization of HESCs by inactivating AKT via the AKT-PP2A complex[24]. This theory was supported by our observation that the knockdown of *ADRA2C* increased AKT phosphorylation and

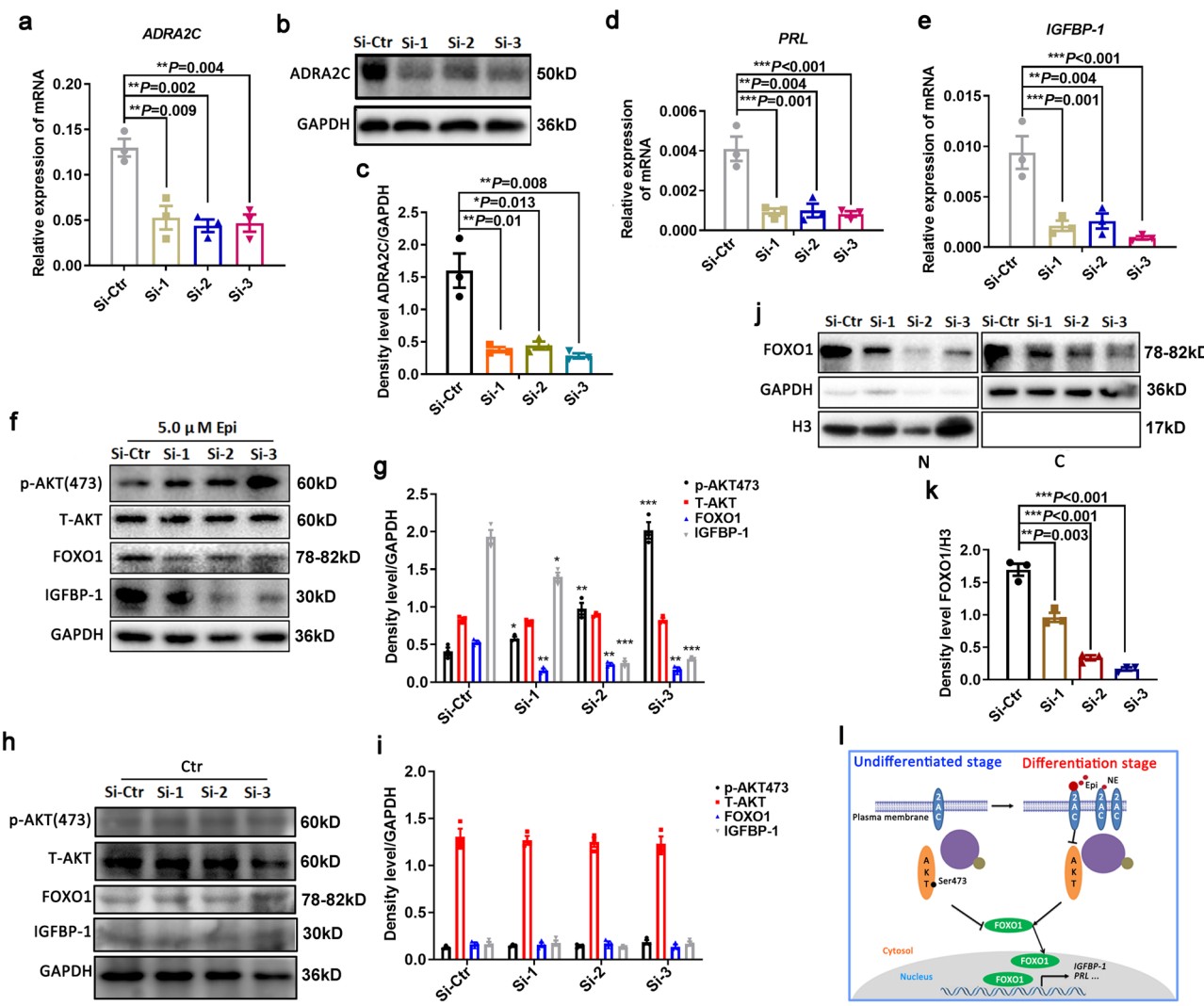

**Fig. 5 Knockdown of ADRA2C abolished the promoting effect of adrenergic signal-mediated decidualization of HESCs. a**, **b** The mRNA and protein expression level for knockdown efficiency of *ADRA2C* in HESCs. **c** Relative fold change in density levels for **b**. **d**, **e** The mRNA level for *PRL* and *IGFBP-1* transcription in the presence of epinephrine after *ADRA2C* knockdown. Values were normalized to *GAPDH* expression level and indicate as the mean ± SEM, *n* = 3. **f** Western blot for the activity of AKT and the expression of FOXO1 in the presence of epinephrine after knockdown of ADRA2C. **g** Relative fold change in density levels for **f**. **h** Western blot for the activity of AKT and the expression of FOXO1 in the absence of epinephrine after knockdown of ADRA2C. **i** Relative fold change in density levels for **h**. **j** Western blot for FOXO1 expression in different cellular compartments in the presence of epinephrine after *ADRA2C* knockdown. **k** Relative fold change in density levels for **j**. **l** Schematic diagram of ADRA2C mediated epinephrine signaling in HESCs during decidualization.

decreased the FOXO1 protein levels in the decidual cells (Fig. 5). FOXO1 was considered an indispensable interacted protein for progesterone receptor during the decidualization and mediated many progesterone response gene transcription[34,35], so the FOXO1 downregulation by the high-level epinephrine treatment may also lead to the progesterone resistant during the decidualization. Although the present findings were somewhat consistent with past results, further experiments were warranted to ascertain these speculations.

To analyze the molecular cascade of events leading to increased disease susceptibility, we employed a stressed mouse model[36]. Stress stimulation can lead to increased serum adrenaline levels, thereby impairing sexual mating and early embryonic development in mice (Fig. 6b–d). Therefore, we analyzed the uterine level of epinephrine in mouse models and found that stress and exogenous epinephrine could upregulate the tissue levels of epinephrine in the uteri (Fig. 7a). Preliminary work has confirmed that *Adra1b, Adra2b,* and *Adrb2* adrenergic receptors are specifically and highly expressed in the mouse decidual cells at early pregnancy. Interestingly, we found that, in mouse models with increased endogenous and exogenous epinephrine, the expressions of *Adra1b, Adra2b,* and *Adrb2* adrenergic receptors in the decidual cells decreased significantly, as also confirmed in the oil-induced artificial decidual model (Fig. 7e, f). Our previous studies have confirmed that maternal epinephrine exposure during early pregnancy inhibits the proliferation of endometrial stromal cells and the key modulators of decidualization, including COX2, BMP2, and WNT4[16]. Hence, it was likely that the decidualization function impaired by exogenous epinephrine was closely related to the aberrant adrenergic receptor signaling in the mouse uteri. Collectively, the in vivo and in vitro experiments have demonstrated that the adrenergic receptor signaling pathway was involved in the physiological and pathological processes of endometrial stromal cells differentiation.

Glucocorticoid (GC) is a stress-induced steroid hormone that is released from the adrenal cortex and is essential for stress

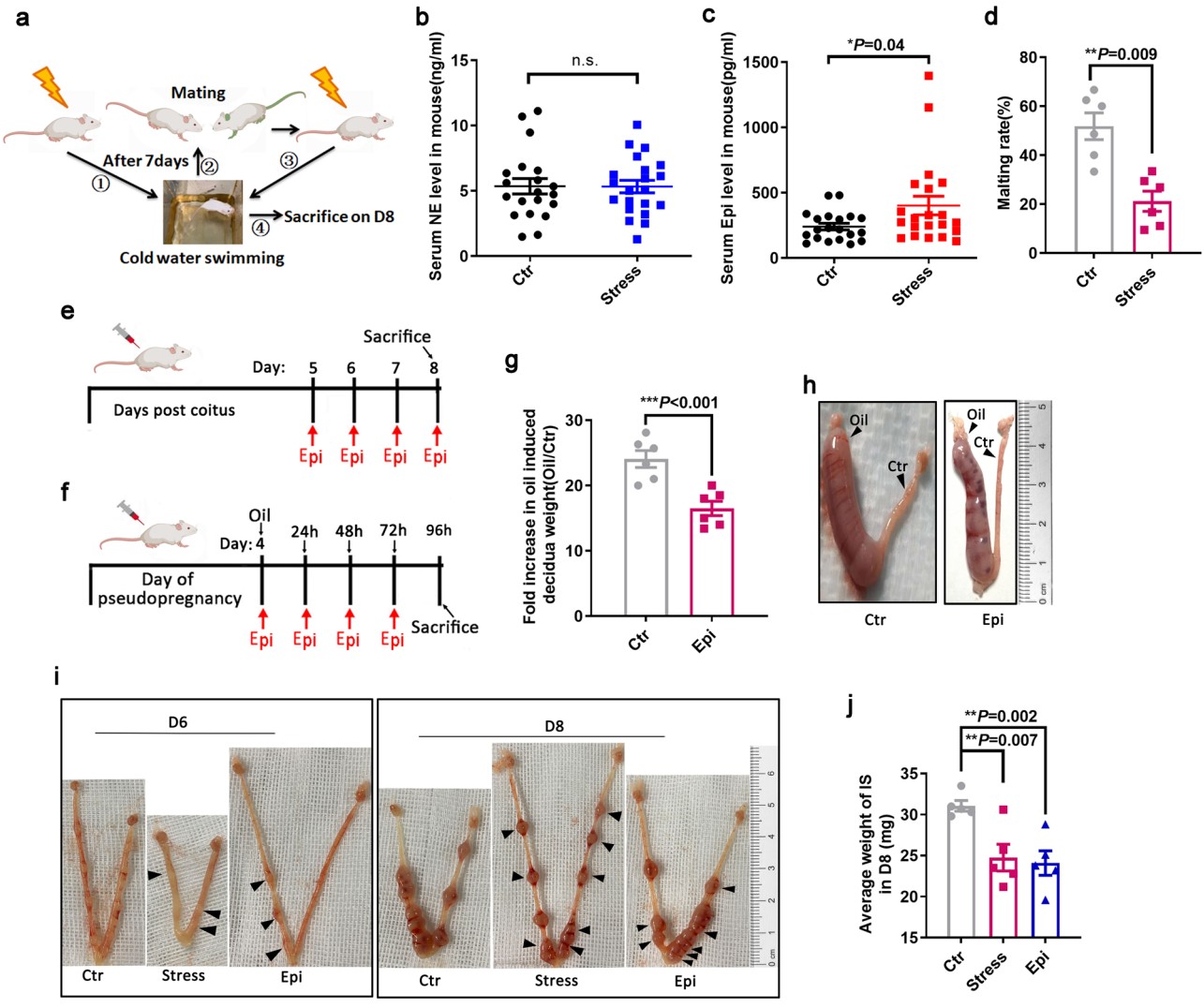

**Fig. 6 Exposure of endogenous and exogenous epinephrine impaired decidua and embryo development. a** Schematic diagram of experimental design of stressed mouse model. **b**, **c** Serum epinephrine and norepinephrine levels in stressed mice ($n = 21$) versus control mice ($n = 20$). **d** Mating rate of stress mice was significantly decreased ($n = 6$) versus control mice ($n = 6$). **e** Schematic diagram of experimental design for exogenous epinephrine exposure. **f** Schematic design of pseudopregnant mice exposed to exogenous epinephrine. **g** The fold increased after oil-induced decidual weight. Ctr, ($n = 6$); Epi, ($n = 6$). **h** Morphological deference of pseudopregnant uterus after 96 h of oil induction between vehicle and 100 μg/mouse epinephrine treatment. Ctr, Non-decidualization ($n = 6$); Oil, artificial decidualization ($n = 6$). **i** At D6 and D8, representative images of intrauterine embryos in mice exposed to exogenous epinephrine and stressed mice were compared with controls. Arrows indicate planting sites with growth restriction. **j** The average weight of IS was decreased in stressed and exogenous epinephrine exposed mice at D8. IS implantation site.

adaptation. GC receptor (GR) in the mouse uterus is essential for normal fertility. The lack of uterine GR signaling often leads to an exaggerated inflammatory response to exuviation, including altered immune cell recruitment[37]. However, stress-induced GCs exposure has an adverse effect on decidualization and uterine receptivity by inducing collagen disorders[38]. These results demonstrate that stress-related hormones and the associated receptors play both physiological functions and pathological damage during early pregnancy. Our data also indicates that the appropriate epinephrine is beneficial to HESCs decidualization, but aberrantly elevated epinephrine has a detrimental effect on early pregnancy.

Taken together, our study offers new insights into the mechanism of adrenergic signaling in human endometrial functioning and provides a firm experimental basis for future research on the relationship between psychological stress factors and early adverse pregnancy outcomes. However, there remain several unsolved problems in this work, warranting further

research for in-depth analyses. For instance, the dynamic changes of ADRA2C during the differentiation process of HESCs decidua need to be ascertained. A proper time course is needed to understand the differences in endometrial stromal cell responses at different ligand concentrations. ADRA2C is highly expressed in the epithelium and stroma but absent in these two compartments of RIF patients. The role of ADRA2C in epithelial cells deserves further investigation. Besides the cell-intrinsic function of stromal expression ADRA2C for decidualization, the epithelial ADRA2C may also be involved in this process through the epithelial-stromal interaction. Psychological anxiety is an extremely complex process. This raises the question of whether any other neurotransmitters are also involved in regulating the functions of the endometrium? Whether abnormal ADRA2C receptor signaling affects PR signaling through FOXO1, leading to the failure of human endometrial receptivity establishment. These unsolved problems are worthy of further research.

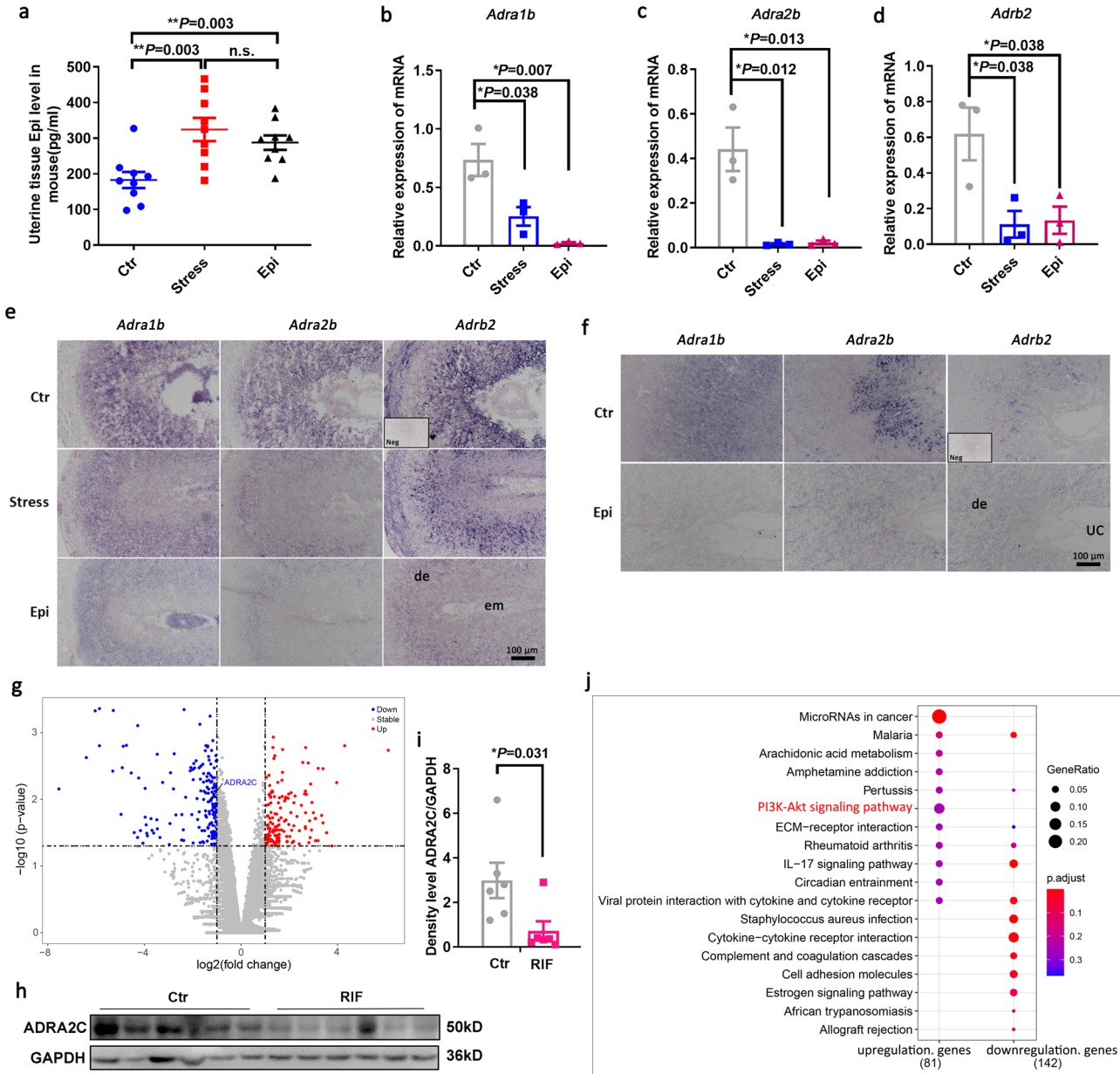

**Fig. 7 High level of epinephrine in vivo may alternate adrenergic receptor expression patterns and signaling pathways in mouse and human endometrial tissues. a** Comparison of epinephrine at D8 endometrial tissues of mice exposed to exogenous epinephrine ($n = 9$) and stressed mice with high serum epinephrine levels ($n = 9$) with control mice ($n = 9$). **b–d** qRT-PCR analyses of *Adra1b*, *Adra2b*, and *Adrb2* at the D8 uteri on stressed mouse and exogenous epinephrine exposed mouse model versus control. **e** In situ hybridization for expression of *Adra1b*, *Adra2b*, and *Adrb2* at the D8 uteri on stressed mouse and exogenous epinephrine exposed mouse model versus control. Scale bars, 100 μm; em embryo, de decidua, Neg negative control. **f** In situ hybridization revealed the alternation expression of *Adra1b*, *Adra2b*, and *Adrb2* at the uteri on 96 h exogenous epinephrine exposed pseudopregnancy mouse model versus control. Scale bars, 100 μm; de decidua, UC uterine cavity, Neg negative control. **g** Volcano plot of the log2 (fold change) and -log10 (*p* value). There were significant differences in RNA expression between RIF and control endometrial receptive tissues (upregulated genes in red and downregulated genes in blue). **h** Western blot analysis of ADRA2C expression in mid-secretory phage (LH+7) endometrial tissues with RIF compared with control. **i** Density levels of ADRA2C in RIF and control for **h**. **j** KEGG pathway analysis of significantly enriched genes and hub gene counts. For each term, the number of enriched genes was indicated by the dot size, while the level of significance was represented by the color. Blue indicates low significance while red represents high significance (adjust *P* < 0.5).

## Methods

**Human subjects**. Blood samples were collected prospectively from 2018 to 2021 from patients who visited the Physical Examination Center and Reproductive Medicine Departments of The Second Affiliated Hospital of Fujian Medical University (Fujian, China). Blood samples were collected in the morning (9:00 AM) from 35 women without fertility concerns for physical examination (age: 25–35 years; control) and from 35 age-matched women with infertility requiring reproductive assistance. Another 69 serum samples were obtained from patients who underwent embryo transfer in the morning of embryo transfer day, and then divided them pregnancy and RIF groups in accordance with the pregnancy outcome. The basic characteristics of the subjects are depicted in Supplementary Tables 1 and 2. A total of 139 human serum samples (70 from the fertility and infertility groups, and 69 from the embryo transfer groups) were employed in the analysis. The levels of epinephrine and norepinephrine were measured via enzyme-linked immunosorbent assay (ELISA) using a commercial kit (Elabscience Biotechnology Co., Wuhan, China). ELISA was performed according to the manufacturer's instructions.

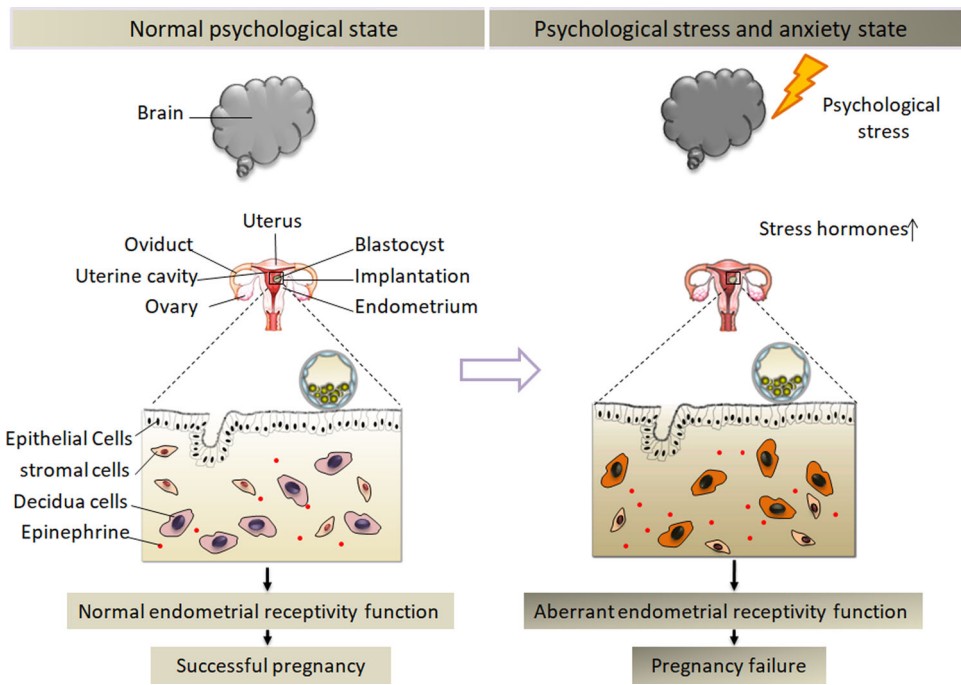

**Fig. 8 Mental stress may alter the functional status of endometrium receptivity, thus affecting early pregnancy.** Normally, the successful implantation and development of embryos depend on the function of the endometrium during the receptivity phase. A certain level of epinephrine is beneficial to the decidual differentiation of human endometrial stromal cells, regulating the receptivity stage of endometrial function. Mental stress can lead to an increased endogenous level of epinephrine. Excessive epinephrine level may interfere with the decidual differentiation of human endometrial stromal cells and thus affect embryo implantation.

**Tissue collection.** This study involved human endometrial biopsy, as approved by The Second Affiliated Hospital of Fujian Medical University. All experiments were conducted in accordance with the Declaration of Helsinki, and all volunteers provided their written informed consent before they participated in the study. Endometrial samples were collected from women with regular menstrual cycles (range: 28–35 days), and no significant intrauterine or ovarian abnormalities were detected on the ultrasonogram. The time of the proliferative phase sample was calculated according to the patient's cycle day, and the time of the luteal phase sample was calculated based on the subject's LH surge and ovulation. After 7 days of LH surge, only the endometrial tissues with no apparent pathology, as assessed by a pathologist, were stored for further experiments. Among these, five subjects who did not have uterine adhesions, inflammation, fibroids, or polyps during hysteroscopy at the mid-stage of proliferation were regarded as control subjects. In addition, seven subjects did not report any reproductive disorders, and the cause of infertility in them was tubal obstruction or male azoospermia, another six subjects reported failure of repeated embryo implantation. No subject received exogenous steroid therapy in the first 3 months. The basic characteristics of the subjects are depicted in Supplementary Table 3. The biopsies for RNA analysis were snap-frozen in liquid nitrogen and stored at −80 °C until further use. The formalin-fixed samples were embedded in paraffin and sectioned for immunohistochemistry. The histological sections were reviewed by a pathologist to verify the histological variants with reference to the criterion of Noyes[39].

**Analysis of differentially expressed mRNAs in the receptivity endometrial tissues.** RNA integrity was assessed using the RNA Nano 6000 Assay Kit of the Bioanalyzer 2100 System (Agilent Technologies, CA, USA). A total amount of 1 µg of RNA per sample was used as the input material for the RNA sample preparations. Briefly, mRNA was purified from total RNA using poly-T oligo-attached magnetic beads. Fragmentation was performed using divalent cations under elevated temperature in the First-Strand Synthesis Reaction Buffer(5X). First-strand cDNA was synthesized with random hexamer primer and M-MuLV Reverse Transcriptase (RNase H-). Second-strand cDNA synthesis was subsequently performed using DNA Polymerase I and RNase H. Remaining overhangs were converted into blunt ends through the exonuclease/polymerase activities. After adenylation of the 3' ends of DNA fragments, adaptors with the hairpin loop structure were ligated to prepare them for hybridization. In order to select cDNA fragments of preferentially 370–420-bp length, the library fragments were purified with the AMPure XP system (Beckman Coulter, Beverly, USA), followed by PCR reaction with the Phusion High-Fidelity DNA polymerase, Universal PCR primers, and Index (X) Primer. Finally, the PCR products were purified (AMPure XP system) and the library quality was assessed on the Agilent Bioanalyzer 2100 system. The clustering of the index-coded samples was performed on the cBot Cluster

Generation System using the TruSeq PE Cluster Kit v3-cBot-HS (Illumina) according to the manufacturer's instructions. After cluster generation, the library preparations were sequenced on the Illumina Novaseq platform to generate 150 bp of paired-end reads.

The differential mRNA expression between the RIF and control endometrial receptivity was evaluated using the edgeR package in R/Bioconductor (version 3.26.5; http://www.bioconductor.org/packages/release/bioc/html/edgeR.html)[40]. We used a bioconductor package complex Heatmap to generate the heatmaps. The DEGs between the data sets were obtained using |log2-fold change| ≥ 0.0, $P \le 0.05$ and |log2-fold change| ≥ 1.0, $P \le 0.05$ as cutoff criteria. To understand the DEGs underlying the biological processes and pathways, GO (geneontology.org) and KEGG (www.genome.jp/kegg) pathway analyses were conducted using the cluster Profiler packages. The GO enrichment results were visualized using the ggplot2 (version 3.2.0; CRAN.R-project.org/package = ggplot2). The KEGG enrichment results were analyzed by the cluster Profiler packages (version 2.1.1; CRAN.R-project.org/package = RSQLite) and org.Hs.eg.db (version 3.8.2; bioconductor.org/packages/org.Hs.eg.db).

**Mouse models.** All experimental mice were housed in a 12-h light/12-h dark cycle (light on 07:00 AM–7:00 PM), with free access to water and food at a temperature of 23 °C. The mice were randomly assigned to groups at the time of purchase. No data sets were excluded from the analyses. These animals were examined daily for any health issues by qualified personnel; the health statuses of all animals were normal. The sample size, sex, and age of all animals used are specified in the respective text and/or figure legends.

**Stress model design.** The stress model test was performed in female mice pre-stressed through forced swimming in water at 15 °C (cold water swimming) for 15 min. The female mice were allowed to swim in a water container (dimension:15-cm diameter, 20-cm height, and 11-cm depth). After swimming, the mice were gently dried by patting the body with a paper towel. The mice were made to repeat swimming thrice a day (from 0800 AM to 1200 AM and at 1600 PM) for 7 consecutive days. On the 7th day, the mice were allowed to mate at 18:00. Virgin female mice were mated with male mice overnight, and, in the morning (08:00 AM), the finding of the vaginal plug was designated as day 1 of pregnancy. The mice were continued to be forced to swim in cold water during the early pregnancy period. The experimental design of the stressed mouse model is shown in Fig. 6a.

**Exogenous epinephrine addition schedules.** The mice were intravenously injected with epinephrine (100 µg/mice) (MedChem Express, Monmouth Junction, USA) in saline at 08:30 AM during days 5–7, once a day, to establish a mouse

model of exogenous epinephrine in the early pregnancy period. The dosage of epinephrine was decided with reference to a previous study[16]. The mice receiving sterile saline injections served as control. The experimental design of the exogenous epinephrine exposure mouse model is depicted in Fig. 6e.

**Artificial decidualization treatment schedules.** Pseudopregnancy was produced by mating females with vasectomized male mice of the same strain (day 1 = vaginal plug). To induce the decidual cell response, sesame oil (25 µL) was infused into a uterine horn on day 4 of pseudopregnancy; while the non-infused contralateral horn served as the control. The mice were injected with epinephrine (100 µg/mice) intraperitoneally 15 min after the intraluminal infusion of oil. Next, the artificially induced decidualization mouse models were injected with epinephrine every 24 h and then sacrificed at 96 h. The animals in the control group received only the vehicle. The uterine weights of the infused and non-infused horns were recorded after 96 h. The animals in the control group received only the vehicle. The experimental design of the artificially induced decidualization mouse model is depicted in Fig. 6f.

**Tissue collection.** The uteri were obtained from the sacrificed mice of models. Before sacrificing, the serums were collected from the stress mouse models. The serum and tissue levels of epinephrine and norepinephrine were determined via ELISA using a mouse kit (Elabscience Biotechnology Co., Wuhan, China). After washing in physiological saline thrice, the implanting sites were frozen and stored in liquid nitrogen until RNA and protein extraction and during the in situ hybridization analyses, respectively. In addition, a portion of tissues was fixed in 4% (wt/vol) paraformaldehyde in 10-mM phosphate-buffered saline (PBS) (pH = 7.4) for histological examination.

**Immortalized HESCs culture and treatments.** The immortalized human endometrial stromal cell line was purchased from the American Type Culture Collection (ATCCRCRL-4003TM) and cultured according to the manufacturer's instructions. HESCs were cultured in phenol red-free DMEM/F-12 (Sigma-Aldrich, USA) containing 10% (vol/vol) charcoal-stripped fetal bovine serum (CS-FBS) (Gibco, Grand Island, USA), 3.1 g/L glucose[41], and 1 mM sodium pyruvate[41], supplemented with 1.5 g/L sodium bicarbonate[41], 50 mg/mL penicillin-streptomycin (Gibco), 1% insulin-transferrin-selenium (Gibco), and 500 ng/mL puromycin[41]. Cell cycle synchronization of HESCs was achieved via serum starvation, and the cells were treated with a medium containing 0.5% CS-FBS for 12 h. After cell starvation, the culture medium was replaced with another medium for further analyses.

The cell proliferation status was measured as a function of 3-(4, 5-dimethylthiazolyl-2)-2, 5-diphenyltetrazolium bromide oxidation (MTS Cell Proliferation Assay Kit) in accordance with the manufacturer's instructions. The proliferation was measured with the optical density at 490 nm on the Synergy-HT Plate Reader (Bio-Tek Instruments). The experiments were performed on three separate occasions. In order to induce stromal cell decidualization in vitro, the cells were treated with a medium containing 2% CS-FBS, 1 mM Medroxy progesterone 17-acetate (MPA; Sigma), and 0.25 mM dibutyryl cAMP (db-cAMP; Sigma), as previously described[42]. The media was refreshed every 48 h, and the cultures were maintained for up to 6 days. In order to study the role of epinephrine-receptor signal in decidualization of stromal cells, we added different concentrations (0.0, 1.0, 2.0, 5.0, and 10.0 µM) of epinephrine into the HESCs in vitro differentiation system and selected the optimal concentration for further analyses. Different concentrations of epinephrine were added simultaneously with the induction of decidualization.

In the RNA interference experiments, 20 mM siRNAs were transfected by Lipofectamine RNAiMAX (Invitrogen, California, USA) according to the manufacturer's instructions, followed by treatment with 5.0 µM epinephrine in the complete medium 6 h after the transfection, and the cells were collected after 96 h for RNA or protein extraction. The sequences of the siRNA were ADRA2C-siRNA target sequence1: 5'-GGATTTCCGGCGATCCTTT-3', ADRA2C-siRNA target sequence2: 5'-TCAACGACGAGACCTGGTA-3', ADRA2C-siRNA target sequence3: 5'-TGTTTTGCACCTCGTCGAT-3'; and control siRNA Med GC was used as the negative control. The same working concentration was used for all three different siRNAs.

**Real-time RT-PCR analysis.** Total RNA was extracted from cultured HESCs by using TRIZOL (Invitrogen) following the manufacturer's protocol. The same input RNA was used for all experiments. The range of RNA concentrations before reverse transcription to cDNA in different experiments was 1–3 mg. The expression levels of different genes were validated by real-time RT-PCR analysis using the ABI 7500 Sequence Detector system according to the manufacturer's instructions (Applied Biosystems, Waltham Massachusetts, USA). All primers used for the real-time PCR in the present study are listed in Supplementary Table 4. The assays were performed at least thrice, each time in duplicate.

**Immunostaining.** The immunohistochemistry analysis was performed as described elsewhere[43]. The specific ADRA2C antibody (1:200; Novus, Colorado, USA) were applied in 5-mm-thick paraffin-embedded sections. The Histostain-SP Kit (Zhong

Shan Golden Bridge Biotechnology, Beijing, China) was used to visualize the antigen. For immunofluorescence, HESCs were fixed with 4% paraformaldehyde[44] in PBS at room temperature for 10 min. After washing with PBS, these cells were permeabilized with 0.2% Triton X-100 in PBS for 10 min and blocked with 0.5% bovine serum albumin for 1 h at room temperature. Anti-ADRA2C (Novus), Ki67 (1:200; Abcam), and FOXO1 (1:200; CST, Danvers, USA) primary antibodies were respectively used in the experiments.

**Separation of nuclear and cytoplasm.** HESCs were collected by cold PBS, resuspended in lysis buffer (10 mM NaCl, 3 mM MgCl2, 0.4% Nonidet P-40, and 10 mM Tris-HCl, pH 7.5) in the presence of protease inhibitor and the nuclei were pelleted, allowing cytoplasm and the nuclei to be isolated for further validation.

**Western blotting.** Protein extraction and western blotting were performed as described previously[43]. ADRA2C (1:500), PRL (1:250; Abcam), IGFBP-1 (1:1000; Abcam), PKA (1:1000; CST, Danvers, USA), p-PKA (1:1000; CST), ERK (1:1000; CST), p-ERK (1:1000; CST), AKT (1:1000; CST), p-Thr (308) AKT (1:1000; CST), p-Ser (473) AKT (1:1000; CST), HOXOA10 (1:1000; Santa Cruz, California, USA), STAT3 (1:1000; CST), FOXO1 (1:1000; CST), GAPDH (1:2000; CST), H3 (1:2000; Abmart, Shanghai, China), and STAT3 (1:1000; CST) were used, respectively. GAPDH and H3 served as the loading control. The bands were visualized using a chemiluminescent substrate (Super Signal West Pico; Thermo Scientific) according to the manufacturer's instructions. The intensity of the bands was determined by using the Quantity One software, and the quantitative analyses of the gray-scale value of each target protein versus that of individual GAPDH were also performed.

**In situ hybridization.** In situ hybridization with digoxygenin was performed as described previously[45]. Briefly, frozen sections (10 µm) were mounted onto poly-L-lysine-coated slides and fixed in 4% paraformaldehyde (Sigma-Aldrich) solution in PBS at 4 °C. After prehybridization, the sections were hybridized at 45 °C for 4 h in a 50% form amide buffer containing digoxygenin-labeled sense or antisense cRNA probes. After hybridization, the sections were incubated with RNase A (20 µg/mL; Takara, Japan) at 37 °C for 20 min and the RNase A-resistant hybrids were incubated with anti-Dig antibody (Roche, Basle Switzerland) and then visualized by NBT/BCIP substrate (Promega, Madison, USA). The nucleus was stained with the Nuclear Fast Red solution (Sigma-Aldrich, N3020). Negative control was performed without a nucleic acid probe.

**Statistical analysis.** All values are depicted as the means ± SEM or mean ± SD of at least three independent experiments. The parametric test was performed. Statistical analysis was performed using the IBM SPSS Statistics 21 program. Independent sample Student's t-test was performed for comparison of the means. The difference was regarded as statistically significant if the two-tailed P value was <0.05.

**Reporting summary.** Experiments on human endometrial stromal cells and mice show that mental stress during early pregnancy can alter the functional status of endometrial receptivity and thereby pregnancy rates. Further information on research design is available in the Nature Research Reporting Summary linked to this article.

## Data availability

RNA sequencing data used in this study are accessible with the following link: https://www.ncbi.nlm.nih.gov/sra/PRJNA838733. All source data underlying the graphs and charts presented in the main figures are presented as Supplementary data. Supplementary data have been uploaded to Figshare and can be accessed through the following link: https://doi.org/10.6084/m9.figshare.20102084. The uncropped blots are available in Supplementary Fig. S5. Element for the mouse in Fig. 6 was created with BioRender.

## Code availability

Details of publicly available software used in the study are given in the "Methods". No custom code or mathematical algorithm that is deemed central to the conclusions was used.

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

## Acknowledgements
This work was supported by the National Natural Science Foundation of China (81901481 to J.W., 81830045 and 82030040 to H.W., 81971388 to S.K.), and Natural Science Foundation of Fujian Province (2020J01016 to S.K.).

## Author contributions
J.W., S.K., and H.W. designed the experiments. J.W. and P.H. performed the experiments. S.L. and L.Q. collected the human endometrial tissues. J.W., Y.J., Y.Z., N.M., and L.W. performed the statistical analysis. J.W., Y.J., M.M., W.D., and Z.L. wrote the manuscript. J.W., C.G., and J.L. revised the manuscript. All authors reviewed the manuscript.

## Competing interests
The authors declare no competing interests.

## Ethical approval
The research was approved by the Biomedical Research Ethics Review Committee of Fujian Medical University ([2019]Fuyi Ethics Review No. (101)) and does not violate ethical principles and agrees to implement the research under the premise of informed consent. According to the requirements of the local ethics committee, the research meets the ethical approval point of view. Informed consent was obtained from all participants who provided blood and endometrial samples. Adult male and female ICR mice (age: 7–8 weeks) were purchased from the Vital River Laboratories Co. Ltd. All animal experiments were approved by the Animal Ethics Committee of Xiamen University and animal protection authorities.
