## [Peer Review File · Communications Biology]

Reviewers' comments:

Reviewer #1 (Remarks to the Author):

The underlying mechanisms of infertility and recurrent pregnancy loss represent a significant knowledge gap in the field of pregnancy biology. A better understanding of those mechanisms may lead to new clinical interventions and improve maternal and fetal health. The current manuscript by Wu et al. builds on their previously published research and utilizes in vitro and in vivo approaches to demonstrate a correlation between adrenergic receptor pathway signaling and aberrant endometrial receptivity and decidualization. The paper will be considerably strengthened by addressing the few points detailed below.

- 1.) Grammatical editing should be performed throughout the manuscript to ensure the accuracy and clarity of the statements made by the authors.
- 2.) In line with the first comment, it is unclear what stage of the cycle each biopsy was taken and if this changes between figures. The authors should explicitly state this information within the figure legends for the reader's reference.
- 3.) Based on the histology presented in Figure 1 and supplemental figure 4, ADRA2C is highly expressed in the epithelium and stroma, and expression is lost in both compartments of RIF patients. Have the authors made an effort to interrogate the role of ADRA2C in the epithelium? Epithelial adrenergic receptor signaling may impact the decidualizing stromal cells in vivo. This should be addressed experimentally in the mouse or addressed in the discussion
- 4.) Although the gel is slightly angled in Figure 2B there is a significant shift in ADRA2C protein size in the last RIF sample presented.
- 5.) As shown in Figure 3, epinephrine increases the expression of IGFBP-1 and PRL on day 4, and knockdown of ADRA2C decreases the expression of those genes (figure 5). If ADRA2C is mediating the effects of exogenous epinephrine, there should be no effect of knockdown on IGFBP-1 and PRL expression during decidualization without epinephrine. That control should be included in figure 5.
- 6.) The reduced level of p-AKT in Figure 4 is not convincing. The authors need to provide quantification of p-AKT /total.
- 7.) The authors state that the expression and distribution patterns of ADRA2C and FOXO1 in RIF patients' endometrium were more heterogeneous than those in controls. However, to this reviewer, the RIF patients appear more homogenous (with one outlier) based on the graphs presented in figure 6. It would also be beneficial to know if the levels of serum epinephrine correlate with ADRA2C and FOXO1 shown in figure 6.
- 8.) The authors should address the role of FOXO1 in progesterone signaling within the discussion. Specifically, is it possible that epinephrine renders the endometrium progesterone resistant?

Reviewer #2 (Remarks to the Author):

This study by Wu and colleagues purports that maternity anxiety impairs embryo implantation through the adverse effects of elevated circulating epinephrine levels on decidual ADRA2C expression. This study is valuable in so far that role of anxiety/stress on embryo implantation and miscarriage rates continues to be a contentious subject in reproduction. Stress is unavoidable in the context of IVF, especially after multiple failures and recurrent pregnancy loss, rendering it difficult to dissect cause

and effect. For example, recurrent pregnancy loss is associated with high stress and depression levels but the level of self-reported emotional distress does not affect the future chance of live birth (PMID: 30819570). Here, the authors report a deceptively simple and plausible mechanism of how circulating stress hormones may impact directly on the implantation process. While the authors draw on various observations, the mechanistic links between these observations are often inferred rather than proven, and further clarifications are required.

1. The authors give the impression that RIF is an established clinical diagnosis - it is not. It is well established that live birth rates in the first IVF cycle are approximately 30% in good prognosis patients. Even after 3 failures, the live birth rate is still above 20% and only drops progressively to 15% in the 9th cycle (see: JAMA. 2015;314(24):2654-2662. doi:10.1001/jama.2015.17296). The high attrition rate at or soon after implantation, in both natural conception cycles or IVF, is not surprising. Human embryos often harbour genetic errors. It is well established that the endometrium engages in embryo biosensing and selection, which means that a large number of implantation failures are caused by a robust physiological process, despite being perceived by both patients and clinicians as 'failures'. The statement on Line 342 that 1/3 of implantation failures can be attributed to poor embryo quality and 2/3 to endometrial factors is pure speculation and not grounded in robust evidence. There are currently no accurate tests to reliably assess embryo quality (or better developmental potential) or to predict endometrial function across different cycles.

2. Other clinical definitions should also be defined clearly. For example, what does 'pregnancy failure' mean in Figure 1C/D? Does this include implantation failures, biochemical losses, and clinical miscarriages? If so, are there differences in epi/NE levels between the different presentations of pregnancy failure? Supplementary Table 2 should include information on the number of previous IVF failures, miscarriages, and successful pregnancies. It is well established that the number of previous failures impact stress levels and, conversely, that a successful pregnancy reduces stress levels.

3. Figure 1C/D: it is unclear from the Methods when the blood samples were obtained. Was this at the time of embryo transfer, some weeks before, at the time of a positive pregnancy test...? Plasma epinephrine (epi) is subjected to significant diurnal rhythm and there is evidence that levels vary across the menstrual cycle - how have the authors controlled for these intrinsic fluctuations?

4. Figure 1C/D: While this figure seems to indicate higher circulating epi levels in the pregnancy failure group, albeit largely overlapping the levels in the pregnancy group, no information is presented on how these discrete differences translate into tissular levels in the endometrium

5. Figure 1E: The authors should show a PCA plot of the samples (control and RIF). The description of exactly how many days after the LH surge these samples were obtained is unclear and not compelling. The authors refer to Noyen's criteria, which are known to be unreliable for timing purposes. Spurious results can easily be obtained by modest timing differences, variation in the cellular composition of the samples, and site of sampling. In endometrial transcriptomics, PCA1 is invariable a time signature and hence will be informative. The authors should also state how many genes were differentially expressed following FDR correction. In addition, a minimal threshold of expression should be used to avoid overinflation. The colour key in this figure is not labelled. GO analysis should be restricted to differentially expressed genes following FDR correction.

6. Figure 1E: What were the circulating epi/NE levels in the samples selected for gene expression profiling? The authors give the impression that RIF equates to higher circulating epi levels and vice versa. That is not what is shown in Fig. 1E.

7. Figure 2A: The data here are poorly presented. It is entirely unclear if genes other than ADRA2C are expressed at all. TPMs would be more informative. Again, the colour key in this figure is not labelled.

8. How were the samples selected for Figure 1B? This figure - on a very limited number of samples - appear to indicate that ADRA2C expression is low in all RIF patients. How is this possible? Did all these patients have elevated epi levels?

9. Single-cell transcriptomic studies demonstrated that ADRA2C is indeed expressed in mid-luteal stromal cells but not in glandular or luminal epithelial cells. Hence, the glandular staining presented in Figure 2C is erroneous and non-specific.

10. Figure 3. Please clarify whether the cultures were decidualized with cAMP/MPA in the presence or absence of increasing doses of epi or solely with epi? It is regrettable that the authors used a cell line instead of primary cultures. Further, recent studies showed that after 4 days of differentiation decidualizing cells diverge into progesterone-dependent decidual cells and progesterone-independent senescent decidual cells that secrete an abundance of inflammatory mediators and ECM modifiers (PMID: 33749894, PMID: 31965050). While PRL and IGFBP are historical decidual markers, they fail to differentiate between decidual populations. Hence, it would be informative if the authors could assess the effect of epi in the subsequent specification of decidual subpopulations. These data may provide important insights into the mechanism of reproductive failure.

11. Figure 4. Downregulation of GPCRs often reflects a feedback mechanism in response to receptor activation. By using a single time point, the authors provide no information on the dynamic changes in ADRA2C across the time course. This is potentially misleading. For example, the difference between 5 and 10 μM of epi treatment on ADRA2C expression may be explained by differences in the kinetics of receptor down-regulation. A proper time course is needed to understand the differences in endometrial stromal cell responses to different ligand concentrations.

12. Figure 6. Again the authors make the unsubstantiated assumption that lower ADRA2C expression is the consequence of higher circulating epi levels. If ADRA2C signalling is important for decidualization, why are low levels of circulating epi levels not associated with reproductive failure? What were the circulating levels and tissue levels in these patients? ADRA2C levels may be lower because of errors in the timing of the samples or a delayed endometrial response.

13. The mice experiments are interesting but not informative as the receptor of interest, i.e., *Adra2c*, appears not to be expressed in murine decidual cells. However, several other receptors are expressed, including *Adra1b*, *Adra2b* and *Adrb2*. The mouse is of course a polytocous species with very different reproductive traits when compared to humans. The lack of evolutionary conservation of adrenergic receptors could indicate that stress signals play a prominent role in murine pregnancy but their role in human pregnancy is dampened or redundant? This is not without precedent. For example, a transient rise in circulating estrogen levels is critical for implantation in mice. While the rise in estrogen levels is conserved in mid-luteal human endometrium, it is no longer required for endometrial receptivity or implantation.

Reviewer #3 (Remarks to the Author):

This is an interesting and very complete manuscript submission. The authors appropriately point out the significant findings but qualify the descriptions as suggesting physiological mechanisms rather than confirming physiological mechanisms.

Following are some concerns that this reviewer would like the authors to address:

1) There are no IgG controls for the immunohistochemistry and Western blots. These controls are essential. Please read Hewitt et al, *J histochem & Cytochem* 2014, 62:693-697.

2) There is no quantification shown for the Western blots. Therefore all talk of increasing or decreasing protein levels is not justified.

3) Did I miss the descriptions for how the nuclear and cytoplasmic fractions were separated and how the in situ hybridization was performed?

4) lines 321-323 - This description is not accurate. NE was not significantly higher, but EP was significantly higher.

5) Where are the controls for the in situ hybridization?

6) Overall the writing is easy to understand, but it could use fine tuning by someone who speaks English as a first language.

March 31th, 2022

Editorial Office,

Collaborative Peer Review Team, Communications Biology

Manuscript ID: COMMSBIO-21-2971A

Dear Dr. Frank Avila,

My co-authors joined me in expressing our sincere appreciation for your time and efforts in handling the above referenced manuscript and to the Editor and reviewers for his/her constructive and thoughtful comments to strengthen the manuscript. We have critically reviewed the comments. All changes in the revised manuscript are highlighted in **Blue**.

General response:

We would like to thank the editor and reviewers for your time and insightful comment. We thank the reviewers for thinking that our manuscript is interesting and timely review. We also learned from your comments that our manuscript still has some weaknesses, in particular, more explanation regarding data collection methods. Based on your suggestions, we have carefully reviewed our manuscript and conformed to your opinions. Below, we have answered the questions of editor and each reviewer.

Reviewers' comments:

Reviewer #1 (Remarks to the Author):

The underlying mechanisms of infertility and recurrent pregnancy loss represent a significant knowledge gap in the field of pregnancy biology. A better understanding of those mechanisms may lead to new clinical interventions and improve maternal and fetal health. The current manuscript by Wu et al. builds on their previously published research and utilizes in vitro and in vivo approaches to demonstrate a correlation between adrenergic receptor pathway signaling and aberrant endometrial receptivity and decidualization. The paper will be considerably strengthened by addressing the few points detailed below.

1.) Grammatical editing should be performed throughout the manuscript to ensure the accuracy and clarity of the statements made by the authors.

Response: Thank you very much for your positive comments. We sincerely thank the reviewer for expressing your concern about the manuscript's English errors. We have invited professional English manuscript editing expert to further revise and improve the manuscript. At the same time, we use a word processing program to check and correct the English spelling errors of our manuscript, and use colored markers to highlight the changes in the manuscript. For professional English manuscript editing experts, see the attachment at the end of the letter.

2.) In line with the first comment, it is unclear what stage of the cycle each biopsy was taken and if this changes between figures. The authors should explicitly state this information within the figure legends for the reader's reference.

Response: Thanks for reviewer's constructive suggestion. We have added corresponding information in the figure legends(**See revised Figure legend 2B and manuscript Line 120-124**).

Line 120-124: *"The time of the proliferative phase sample was calculated according to the patient's cycle day, and the time of the luteal phase sample was calculated based on the subject's LH surge and ovulation. After 7days of LH surge, only the endometrial tissues with no apparent pathology, as assessed by a pathologist, were stored for further experiments."*

3.) Based on the histology presented in Figure1 and supplemental figure 4, ADRA2C is highly expressed in the epithelium and stroma, and expression is lost in both compartments of RIF patients. Have the authors made an effort to interrogate the role of ADRA2C in the epithelium? Epithelial adrenergic receptor signaling may impact the decidualizing stromal cells in vivo. This should be addressed experimentally in the mouse or addressed in the discussion

Response: Thanks for this concern to strength our manuscript and we have addressed more new insights in the discussion(**See detail information in revised**

manuscript Line 520-524, Line 594-598).

Line 520-524: "Furthermore, we observed that ADRA2C signaling was highly present in both the endometrial epithelia and stroma, and that the expression was lost in both the compartments of RIF patients. This observation suggested that adrenergic receptor signaling might exhibit biological functions in both epithelial and stromal differentiation."

Line 594-598: "ADRA2C is highly expressed in the epithelium and stroma but absent in these two compartments of RIF patients. The role of ADRA2C in epithelial cells deserves further investigation. Besides the cell-intrinsic function of stromal expression ADRA2C for decidualization, the epithelial ADRA2C may also be involved in this process through the epithelial-stromal interaction."

4.) Although the gel is slightly angled in Figure 2B there is a significant shift in ADRA2C protein size in the last RIF sample presented.

Response: Thanks for this concern. It may be that the abnormality of the WB system caused the gel to tilt slightly in Figure 2B, resulting in a significant change in the size of the ADRA2C protein. Since the information suggested by Figure 2B data is the same as Figure 7H, the data in Figure 2B was removed (**See detail information in revised Figure 2 and 7**).

5.) As shown in Figure 3, epinephrine increases the expression of IGFBP-1 and PRL on day 4, and knockdown of ADRA2C decreases the expression of those genes (figure 5). If ADRA2C is mediating the effects of exogenous epinephrine, there should be no effect of knockdown on IGFBP-1 and PRL expression during decidualization without epinephrine. That control should be included in figure 5.

Response: Many thank for this constructive suggestion to strength our manuscript. We agree with the reviewer's point of view. The new control experiments were performed, and just as the reviewer proposed, knockdown of ADRA2C had no effect on IGFBP-1 and PRL expression in the absence of epinephrine, validating the role of ADRA2C in mediating exogenous epinephrine (**See detail information in**

revised Figure 5H and I).

6.) The reduced level of p-AKT in Figure 4 is not convincing. The authors need to provide quantification of p-AKT /total.

Response: As suggestion, we have provided quantification of p-AKT /total in Figure 4(*See detail information in revised Figure 4B*).

7.) The authors state that the expression and distribution patterns of ADRA2C and FOXO1 in RIF patients' endometrium were more heterogeneous than those in controls. However, to this reviewer, the RIF patients appear more homogenous (with one outlier) based on the graphs presented in figure 6. It would also be beneficial to know if the levels of serum epinephrine correlate with ADRA2C and FOXO1 shown in figure 6.

Response: According to your comment, we have carefully checked the Figure 6. This is true, and due to insufficient evidence, this is a real problem causing divergent views. We apologize for the imprecise conclusion, and we deleted the relevant information in the figures and statement in manuscript(*See detail information in revised Figure 7H-I*).

Regrettably, no corresponding serum was retained for endometrial samples. To make up for this shortcoming, we detected serum epinephrine levels in RIF patients, indirectly proved the correlation between serum epinephrine levels and ADRA2C and FOXO1. The levels of serum epinephrine in RIF patients shown in revised figure 1C and D(*See detail information in revised Figure 1C-D and Figure 7A*).

8.) The authors should address the role of FOXO1 in progesterone signaling within the discussion. Specifically, is it possible that epinephrine renders the endometrium progesterone resistant?

Response: Progesterone signaling is important in the establishment of the endometrial receptive state and the decidualization of HESCs. We fully agree with reviewer's opinion that the role of FOXO1 in progesterone signaling should be

addressed in the discussion. We have made corresponding changes in our revised manuscript (*See detail information in revised manuscript Line 548-553*).

Line 548-553: "FOXO1 was considered as an indispensable interacted protein for PGR during the decidualization (PMID:11893744) and mediated many progesterone response gene transcription (PMID:25584414), so the FOXO1 downregulation by the high level epinephrine treatment may also lead to the progesterone resistant during the decidualization."

Reviewer #2 (Remarks to the Author):

This study by Wu and colleagues purports that maternity anxiety impairs embryo implantation through the adverse effects of elevated circulating epinephrine levels on decidual ADRA2C expression. This study is valuable in so far that role of anxiety/stress on embryo implantation and miscarriage rates continues to be a contentious subject in reproduction. Stress is unavoidable in the context of IVF, especially after multiple failures and recurrent pregnancy loss, rendering it difficult to dissect cause and effect. For example, recurrent pregnancy loss is associated with high stress and depression levels but the level of self-reported emotional distress does not affect the future chance of live birth (PMID: 30819570). Here, the authors report a deceptively simple and plausible mechanism of how circulating stress hormones may impact directly on the implantation process. While the authors draw on various observations, the mechanistic links between these observations are often inferred rather than proven, and further clarifications are required.

Response: Thank you very much for your time and effort to review our manuscripts, and consider this study is valuable. My co-authors and I would like to express our heartfelt thanks for your thoughtful comments on improving the quality of this manuscript. We have carefully considered these your comments, and hereby respond to the comments point by point in our revised manuscript.

1. The authors give the impression that RIF is an established clinical diagnosis - it is not. It is well established that live birth rates in the first IVF cycle are approximately

30% in good prognosis patients. Even after 3 failures, the live birth rate is still above 20% and only drops progressively to 15% in the 9th cycle (see: JAMA. 2015;314(24):2654-2662. doi:10.1001/jama.2015.17296). The high attrition rate at or soon after implantation, in both natural conception cycles or IVF, is not surprising. Human embryos often harbour genetic errors. It is well established that the endometrium engages in embryo biosensing and selection, which means that a large number of implantation failures are caused by a robust physiological process, despite being perceived by both patients and clinicians as 'failures'. The statement on Line 342 that 1/3 of implantation failures can be attributed to poor embryo quality and 2/3 to endometrial factors is pure speculation and not grounded in robust evidence. There are currently no accurate tests to reliably assess embryo quality (or better developmental potential) or to predict endometrial function across different cycles.

Response: We agree with reviewer's comments. Indeed, the etiology of RIF is very complex, and even the result of multiple factors. There are currently no accurate tests to predict endometrial function across different cycles. There is no clear evidence for the statement that 1/3 of implantation failures are attributable to poor embryo quality and 2/3 to endometrial factors. Thereby, we remove speculation without robust evidence (*See detail information in revised manuscript Line 346*).

Line 346: ".....We then assessed whether adrenergic receptor signaling was involved in....."

2. Other clinical definitions should also be defined clearly. For example, what does 'pregnancy failure' mean in Figure 1C/D? Does this include implantation failures, biochemical losses, and clinical miscarriages? If so, are there differences in epi/NE levels between the different presentations of pregnancy failure? Supplementary Table 2 should include information on the number of previous IVF failures, miscarriages, and successful pregnancies. It is well established that the number of previous failures impact stress levels and, conversely, that a successful pregnancy reduces stress levels.

Response: We agree with this constructive comment. In the current research, the 'pregnancy failure' in Figure 1C/D includes implantation failures, biochemical

losses, and clinical miscarriages. To eliminate interference from other types of 'pregnancy failure', we collected a new batch of samples and only included serum data from patients with RIF. We have made corresponding modification to the revised manuscript(*See detail information in revised Figure 1C – D and manuscript Line 103-106*).

Line 103-106: "Another 69 serum samples were obtained from patients who underwent embryo transfer in the morning of embryo transfer day, and then divided the pregnancy and RIF groups in accordance with the pregnancy outcome."

3. Figure 1C/D: it is unclear from the Methods when the blood samples were obtained. Was this at the time of embryo transfer, some weeks before, at the time of a positive pregnancy test...? Plasma epinephrine (epi) is subjected to significant diurnal rhythm and there is evidence that levels vary across the menstrual cycle – how have the authors controlled for these intrinsic fluctuations?

Response: That's a very good point of view. The blood samples were obtained at the time of embryo transfer. It is true that plasma epinephrine (Epi) is significantly affected by circadian rhythm and menstrual cycle. That's something we have very little control over. In order to eliminate the interference of endogenous rhythm and cycle, we collected serum samples from patients who underwent physical examination at 9:00 am. For embryo transfer patients, we collected serum samples in the morning of embryo transfer day. We have added this detail in the Methods chapter of our revised manuscript (*See detail information in revised manuscript Line 101-106*).

Line 101-106: "Blood samples were collected in the morning (9:00 AM) from 35 women without fertility concerns as physical examination (age: 25–35 years; control) and from 35 age-matched women with infertility requiring reproductive assistance. Another 69 serum samples were obtained from patients who underwent embryo transfer in the morning of embryo transfer day, and then divided them pregnancy and RIF groups in accordance with the pregnancy outcome."

4. Figure 1C/D: While this figure seems to indicate higher circulating epi levels in the pregnancy failure group, albeit largely overlapping the levels in the pregnancy group, no information is presented on how these discrete differences translate into tissular levels in the endometrium.

Response: This is a valuable and constructive comment. Unfortunately, serum and endometrial samples from these patients were not collected at the same time, so data for this analysis are lacking. To make up for this shortcoming, we analyzed the uterine level of epinephrine in mouse models, and found that stress can upregulated the tissue level of epinephrine in the uteri. We have added new insights into the manuscript(*See detail information in revised Figure 7A and manuscript Line 472-476*).

Line 472-476: "Primarily, we found that mice with high serum epinephrine levels corresponded to higher epinephrine in the uterine tissues when compared to the controls (Fig. 7A). This means that the level of circulating epinephrine fluctuates in a similar way to the level of epinephrine in the uterine tissue."

5. Figure 1E: The authors should show a PCA plot of the samples (control and RIF). The description of exactly how many days after the LH surge these samples were obtained is unclear and not compelling. The authors refer to Noyen's criteria, which are known to be unreliable for timing purposes. Spurious results can easily be obtained by modest timing differences, variation in the cellular composition of the samples, and site of sampling. In endometrial transcriptomics, PCA1 is invariably a time signature and hence will be informative. The authors should also state how many genes were differentially expressed following FDR correction. In addition, a minimal threshold of expression should be used to avoid overinflation. The colour key in this figure is not labelled. GO analysis should be restricted to differentially expressed genes following FDR correction.

Response: We fully agree with reviewer's point and we have provided a PCA plot of the samples in supplemental Figure S1. The description of the days after the LH surge these samples were obtained is provided in the legend of Figure 2B(*See detail*

information in revised Figure 2B and revised manuscript Line 122-124).

The comment about GO analysis is a very professional and important. Thank you for your attention to this regard. However, the number of differentially expressed genes after FDR correction is not very large, so the automatic parameter adjustment is $p\text{-value} \leq 0.05$ during the process analysis. The number of differentially expressed genes after parameter adjustment has been supplemented (**See detail information in revised Figure 1E and supplemental Figure S1**).

Line 122-124: *“After 7 days of LH surge, only the endometrial tissues with no apparent pathology, as assessed by a pathologist, were stored for further experiments.”*

6. Figure 1E: What were the circulating epi/NE levels in the samples selected for gene expression profiling? The authors give the impression that RIF equates to higher circulating epi levels and vice versa. That is not what is shown in Fig. 1E.

Response: That is a very good point of view. Unfortunately, the patient's serum epinephrine was not tested at the time of endometrial collection, which is a regrettable oversight. In order to address this shortcoming, we analyzed the level of Epi/NE levels in serum and uterine tissue of the mouse models. The relevant data are provided in Figure 7A and B (**See detail information in revised Figure 7A and manuscript Line 472-476**).

Line 472-476: *“Primarily, we found that mice with high serum epinephrine levels corresponded to higher epinephrine in the uterine tissues when compared to the controls (Fig. 7A). This means that the level of circulating epinephrine fluctuates in a similar way to the level of epinephrine in the uterine tissue.”*

7. Figure 2A: The data here are poorly presented. It is entirely unclear if genes other than ADRA2C are expressed at all. TPMs would be more informative. Again, the colour key in this figure is not labelled.

Response: Thanks reviewer for your sincere suggestions. We have made corresponding changes in our revised figure (**See detail information in revised Figure**

2A).

8. How were the samples selected for Figure 2B? This figure - on a very limited number of samples - appear to indicate that ADRA2C expression is low in all RIF patients. How is this possible? Did all these patients have elevated epi levels?

Response: We appreciate the reviewer's helpful point of view. In fact, we found that endometrial ADRA2C was reduced in the majority of these RIF patients. Here, we selected a subset of patients based on the preliminary data to highlight the under-expression of ADRA2C in RIF. Same with the concern 6, we didn't collect the matched serum and endometrium sample at the same time and now we analyzed the serum and tissue Epi level in the stressed mouse model. It was uncovered that both the serum and tissue level of Epi was increased, accompanied with the reduced uterine receptor expression in the stressed mice. Since the information suggested by the data in Figure 2B is the same as that in Figure 6C, we remove Figure 2B (*See detail information in revised Figure 2 and 7*).

9. Single-cell transcriptomic studies demonstrated that ADRA2C is indeed expressed in mid-luteal stromal cells but not in glandular or luminal epithelial cells. Hence, the glandular staining presented in Figure 2C is erroneous and non-specific.

Response: Thanks for this concern. In fact, we also doubted the specificity of the ADRA2C antibody. However, during the experiment, we have used ADRA2C antibodies from different manufacturers, and the results are consistent. Single-cell transcriptomic studies demonstrated that ADRA2C is not expressed in glandular or luminal epithelial cells, but the protein was detectable in this study. We think the possible reasons are: 1. ADRA2C protein expression is relatively stable in the epithelial cells, and mRNA may not be required; 2. The detect limitation of analyzing the ADRA2C expression through single-cell transcriptional sequencing; 3. Using RT-PCR, we did detect ARDA2C expression in uterine epithelial organoids, through the mRNA level seems low compared with that in stromal cells. The result is shown in the following figure just for the reviewers.

10. Figure 3. Please clarify whether the cultures were decidualized with cAMP/MPA in the presence or absence of increasing doses of epi or solely with epi? It is regrettable that the authors used a cell line instead of primary cultures. Further, recent studies showed that after 4 days of differentiation decidualizing cells diverge into progesterone-dependent decidual cells and progesterone-independent senescent decidual cells that secrete an abundance of inflammatory mediators and ECM modifiers (PMID: 33749894, PMID: 31965050). While PRL and IGFBP are historical decidual markers, they fail to differentiate between decidual populations. Hence, it would be informative if the authors could assess the effect of epi in the subsequent specification of decidual subpopulations. These data may provide important insights into the mechanism of reproductive failure.

Response: We agree with that this is an interesting question. In the process of exploring the optimal concentration of Epi in this study, we have confirmed that Epi alone cannot stimulate decidualization of HESCs in the absence of cAMP, probably due to lack of adequate ADRA2C expression. In addition, we supplemented the experiment to analyze the differential expression of *DIO2* and *SACRA5*, which represent the two subpopulations in decidualized HESCs under Epi stimulation. Epi didn't show an obvious effect on the specification of decidual subpopulations based on the ratio of *DIO2* and *SACRA5* expression level. Since this is a complex scientific question requiring more experimental validation, the data are not presented in revised manuscript. The results are shown in the following figures just for the reviewers.

11. Figure 4. Downregulation of GPCRs often reflects a feedback mechanism in response to receptor activation. By using a single time point, the authors provide no information on the dynamic changes in ADRA2C across the time course. This is potentially misleading. For example, the difference between 5 and 10 μM of epi treatment on ADRA2C expression may be explained by differences in the kinetics of receptor down-regulation. A proper time course is needed to understand the differences in endometrial stromal cell responses to different ligand concentrations.

Response: Stimulation of GPCRs by its ligand responds relatively quickly, sometimes even within minutes(PMID: 34847077). We totally agree the point that a proper time course is needed to understand the differences in endometrial stromal cell responses to different ligand concentrations. In this study, we found high serum concentrations of Epi appear to be detrimental to successful pregnancy. This cellular experiment was designed to reveal the molecular alterations involved in this phenotype. We explored the appropriate timing and dose of Epi effects to explore the corresponding changes in the involved molecules. We begged the reviewer's agreement that our research is not a molecular biology study, so no in-depth biological exploration has been done in this work. However, this point of view is very valuable and interesting, and worthy of further study in the future.

12. Figure 6. Again the authors make the unsubstantiated assumption that lower ADRA2C expression is the consequence of higher circulating epi levels. If ADRA2C signalling is important for decidualization, why are low levels of circulating epi levels not associated with reproductive failure? What were the circulating levels and tissue levels in these patients? ADRA2C levels may be lower because of errors in the timing

of the samples or a delayed endometrial response.

Response: That is very constructive point of view. We agree with this valuable suggestion. It is regrettable that serum samples and endometrium from our patients were not collected at the same time, so data for this analysis are lacking. To make up for this shortcoming, we analyzed in vivo experiments in mouse models. We have added new insights into the manuscript (*See detail information in revised Figure 7A and manuscript Line 472-476*).

Line 472-476: "Primarily, we found that mice with high serum epinephrine levels corresponded to higher epinephrine in the uterine tissues when compared to the controls (Fig. 7A). This means that the level of circulating epinephrine fluctuates in a similar way to the level of epinephrine in the uterine tissue."

13. The mice experiments are interesting but not informative as the receptor of interest, i.e., Adra2c, appears not to be expressed in murine decidual cells. However, several other receptors are expressed, including Adra1b, Adra2b and Adrb2. The mouse is of course a polytocous species with very different reproductive traits when compared to humans. The lack of evolutionary conservation of adrenergic receptors could indicate that stress signals play a prominent role in murine pregnancy but their role in human pregnancy is dampened or redundant? This is not without precedent. For example, a transient rise in circulating estrogen levels is critical for implantation in mice. While the rise in estrogen levels is conserved in mid-luteal human endometrium, it is no longer required for endometrial receptivity or implantation.

Response: We are pleased that reviewer found mouse experiments were interesting. Indeed, many evidences from the mouse studies have provided the valuable clue for our understanding the regulation occurred in humans, even though it may be not the same gene or molecular. Although human and mouse endometrial stromal cells have different subtypes of adrenergic receptors involved in the differentiation of endometrial stromal cells, both have phenotypes that are unfavorable for embryonic development after being stimulated by high concentrations of Epi. The mechanism for the regulated expression of adrenergic

receptors in the uterine cell may be different between the human and mice, which could be due to the different regulatory element in their genomes, but the receptor response to the ligand and the downstream effects were conserved between the human and mice.

Reviewer #3 (Remarks to the Author):

This is an interesting and very complete manuscript submission. The authors appropriately point out the significant findings but qualify the descriptions as suggesting physiological mechanisms rather than confirming physiological mechanisms.

Response: Thank you very much for your positive comments. We sincerely thank the reviewer for thinking the manuscript interesting and very complete. Thanks for reviewer's constructive suggestion. We have critically reviewed the comments and added more new insights and suggestions in the manuscript.

Following are some concerns that this reviewer would like the authors to address:

1) There are no IgG controls for the immunohistochemistry and Western blots. These controls are essential. Please read Hewitt et al, J histochem&Cytochem 2014, 62:693-697.

Response: We are very grateful for this constructive suggestion to enhance the quality of our manuscripts. We fully agree with the reviewer's opinion. In fact, each of our experiments has a corresponding control experiment, but it is not shown in the figures. We apologize for the confusion caused. As suggested, we have added the control information to the corresponding figures (***See detail information in revised Figure 2B, 4G, S2C, S3B and S4***).

2) There is no quantification shown for the Western blots. Therefore all talk of increasing or decreasing protein levels is not justified.

Response: Thanks for reviewer's constructive suggestion. We have added quantification for the Western blots in the manuscript (***See detail information in***

revised Figure 4, 5 and 7).

3) Did I miss the descriptions for how the nuclear and cytoplasmic fractions were separated and how the in situ hybridization was performed?

Response: Based on your comments, we have carefully checked the relevant chapters of methods. We found that the description of the nuclear and cytoplasmic fractions and in situ hybridization were indeed confusing. We apologize for the confusion caused. Therefore, we have added the details of the relevant chapters of methods to the revised manuscript (**See detail information in revised manuscript Line 278-282 and Line 296-307**).

Line 278-282: *“Separation of Nuclear and Cytoplasm HESCs were collected by cold PBS, resuspended in lysis buffer (10 mM NaCl, 3 mM MgCl₂, 0.4% Nonidet P-40, and 10 mM Tris-HCl, pH 7.5) in the presence of protease inhibitor and the nuclei were pelleted, allowing cytoplasm and the nuclei to be isolated for further validation.”*

Line 296-307: *“In situ hybridization In situ hybridization with digoxigenin was performed as described previously [23]. Briefly, frozen sections (10 μm) were mounted onto poly-L-lysine coated slides and fixed in 4% paraformaldehyde (Sigma-Aldrich) solution in PBS at 4 °C. After prehybridization, the sections were hybridized at 45 °C for 4 h in a 50% formamide buffer containing digoxigenin-labeled sense or antisense cRNA probes. After hybridization, the sections were incubated with RNase A (20 μg/mL; Takara, Japan) at 37 °C for 20 min and the RNase A-resistant hybrids were incubated with anti-Dig antibody (Roche, Basle Switzerland) and then visualized by NBT/BCIP substrate (Promega, Madison, USA). The nucleus was stained with the NuclearFast Red solution (Sigma-Aldrich, N3020). Negative control was performed without nucleic acid probe.”*

4) lines 321-323 - This description is not accurate. NE was not significantly higher, but EP was significantly higher.

Response: After a careful search of the chapter and figures, we did find the

misunderstood areas. This is a serious criticism and we apologize for this mistake. We have added this modification in the revised manuscript(***See detail information in revised manuscript Line 322-325***).

Line 322-325:“However, in patients with infertility who underwent embryo transfer, only the serum epinephrine levels of those with failed embryo implantation were significantly higher than those with pregnancy. (Fig. 1C-D).”

5) Where are the controls for the in situ hybridization?

Response: Thanks for this concern to strength our manuscript. We very much agree with the sincere concerns. Sorry for not providing this information. We have made corresponding additions in the revised manuscript and figures (***See detail information in revised Figure 7E-F***).

6) Overall the writing is easy to understand, but it could use fine tuning by someone who speaks English as a first language.

Response: Thank you very much for your positive comments. We accept review’s comment for expressing your concern about the manuscript’s English errors. We have invited professional English manuscript editing expert to further revise and improve the manuscript. At the same time, we use a word processing program to check and correct the English spelling errors of our manuscript, and use colored markers to highlight the changes in the manuscript. For professional English manuscript editing experts, see the attachment below.

<https://www.mjeditor.com>

EDITORIAL CERTIFICATE

This document certifies that the manuscript listed below was edited for grammar, punctuation, spelling, and overall style by one or more expert native English speaking editors with a PhD degree.

Manuscript information

ID: MJ15470

Editing date: 2022.03.25

Title :Maternal anxiety affects embryo implantation via impairing adrenergic receptor signaling in decidual cells.

Author :Jinxiang Wu, Shu Lin, Pinxiu Huang, Lingling Qiu, Yufei Jiang, Ying Zhang, Nan Meng, Meiqing Meng, Lemeng Wang, Wenbo Deng, Zhao Liu, Chuanhui Guo, Jinhua Lu, Haibin Wang, Shuangbo Kong.

Language writing before editing: Very poor Poor Fair Good Very good Excellent

Recommendation after language editing: Submitting to target journal directly
Submitting to target journal after minor revision
Re-editing required after major revision
Not suitable for publication

Certificate by

Editor in Chief
MJ Language Editing Services, Shenzhen, China

Disclaimer:Our service does not involve authenticity review or ethical review on the data (including images) presented in the manuscript. Neither the research content nor the author's intentions were altered in any way during the editing process. Documents receiving this certification should be English-ready for publication. The authors have the option to accept or reject our suggestions and changes in the edited document. However, we do not bear responsibility for revisions made to the document after our editing. If the manuscript is suspected of plagiarism, please contact the authors in time.

MJ Language Editing Services
Diwang Building, No. 5002 Shennan Road, Luohu District, Shenzhen, China
Tel:+086 0755 25100506

REVIEWERS' COMMENTS:

Reviewer #1 (Remarks to the Author):

The authors have adequately addressed my previous comments.

Reviewer #2 (Remarks to the Author):

The authors have demonstrably improved their manuscript, although two outstanding issues remain:

1. The PCA plot provided in the new Figure S1 is informative. As mentioned before, PC1 invariably indicates the molecular time in endometrial transcriptomics, reflecting the conspicuous temporal gene changes in cycling endometrium. Clearly, the samples segregate in PC1, although not between control and RIF. However, perfect separation of control and RIF samples was achieved in PC2, accounting for 19% of the variance in gene expression. This is to a certain extent unexpected as some women labelled RIF are expected to have successful future pregnancies. The clear separation of case versus control samples in PC2 raises the question as to why so few genes are differentially expressed following multiple testing corrections? It may be the case that the time signature (PC1) obscures genuinely DEG between the clinical groups. This issue has previously been addressed by several investigators in the field and several bioinformatics tools for time correction have been developed. See methods in PMID: 35085395 and in PMID: 35092277. There is a good reason to insist on multiple testing corrections as the chance of spurious and unverifiable results is high with this type of data. Further, time correction may lead to the discovery of genes initially deemed not differentially expressed.
2. I do not understand the authors' rejection of my request for time-course information on ADRA2C expression because 'this is not a molecular biology study'? This is a straightforward experiment that will aid the reader to understand the relationship between ADRA2C expression and activity.
3. L460: These interesting findings suggest that elevated epinephrine levels in vivo disrupt the decidualization of uterine stromal cells, leading to early embryo development. Do the authors mean embryo 'development' or 'demise'?
4. The Discussion is brief and to the point. However, what is unclear is how the authors envisage the results of this study to impact the clinical management of RIF patients? Do they call for a clinical trial or routine screening of circulating Epi levels prior to IVF?

Reviewer #3 (Remarks to the Author):

The authors have satisfied this reviewers concerns.